# On the Relationship Between Activation Outliers and Feature Death in Sparse Autoencoders

Elana Simon [1]    Etowah Adams [2]    James Zou [1]

## Abstract

Sparse autoencoders (SAEs) decompose neural network activations into interpretable features, but many learned features never activate, a problem called feature death that wastes dictionary capacity and can reintroduce superposition. Death rates vary dramatically between models: near-zero on GPT-2, over 70% on AlphaFold3 with identical configurations. We find that dimension-level activation outliers (dimensions whose mean magnitude is large relative to per-token variation) cause this by shifting pre-activations at initialization based on each feature's alignment with the activation mean. Features anti-aligned with the mean receive permanently negative pre-activations and never fire. We formalize outlier severity as $\gamma = \|\boldsymbol{\mu}\|/\|\boldsymbol{\sigma}\|$; it predicts initial death rates (Spearman $\rho = 0.89$ for dead-by-TopK, 0.82 for dead-by-ReLU) across 454 model-layer combinations spanning language, vision, protein, and genomic models. Dead features can revive during training, but recovery requires the SAE bias to learn the activation mean, a process that is prohibitively slow at high $\gamma$. Mean-centering (subtracting the activation mean) sidesteps this and eliminates outlier-induced death across all tested models, confirming the mechanism and providing a principled basis for when and why this preprocessing step is necessary.

## 1. Introduction

Identical sparse autoencoders (same architecture, same hyperparameters, same auxiliary losses) produce near-zero dead features on GPT-2 (Radford et al., 2019) and 72% dead features on AlphaFold3 (Abramson et al., 2024). Why?

[1]Stanford University [2]Columbia University. Correspondence to: James Zou <jamesz@stanford.edu>.

*Proceedings of the $43^{rd}$ International Conference on Machine Learning*, Seoul, South Korea. PMLR 306, 2026. Copyright 2026 by the author(s).

Neural networks are thought to represent more concepts than they have dimensions, encoding them as overlapping directions in activation space, a phenomenon called superposition (Elhage et al., 2022). Sparse autoencoders (SAEs) attempt to reverse this: they map activations to a higher-dimensional space and enforce sparsity, learning a dictionary of directions (Bricken et al., 2023b; Huben et al., 2024). Each direction, or *feature*, ideally corresponds to a single interpretable concept; features that never activate on any input are called *dead features*. With 70% of features dead, a 32k-element dictionary represents at most 9.6k concepts, and the surviving features must compensate, potentially reintroducing the superposition the SAE was meant to resolve. Worse, death rates vary unpredictably across models and layers (20% to 80% within ESM3 alone), making the effective dictionary size difficult to control.

Several techniques attempt to revive dead features by providing gradient signal to dictionary elements that would otherwise receive none (Jermyn & Templeton, 2024; Gao et al., 2024; Bricken et al., 2023b). These help on some models but, as we show, remain ineffective where death is most severe.

The pattern does not reduce to model family or domain: not all protein models have high death, and even within ESM3 (Hayes et al., 2025), death rates range from 20% to 80% across layers with the same SAE configuration. Whatever causes death lives in the activations themselves, varying not just across models but layer by layer within each model.

The cause is what we call *dimension-level activation outliers*. In layers with high death, certain activation dimensions have mean values that are large relative to their per-token variation, creating a near-constant offset across all inputs. These differ from the token-level outliers studied in quantization contexts (Sun et al., 2024; Dettmers et al., 2022): token-level outliers are properties of individual inputs, while the outliers we study are properties of dimensions, persistent across all inputs (per-model visualizations in Figure S1). These outliers cause dead features through a geometric mechanism visible at initialization, before any training occurs. Each feature's pre-activation (the value computed before the sparsity nonlinearity) decomposes into a constant term, the feature's alignment with the activation mean $\boldsymbol{\mu}$, and a varying term

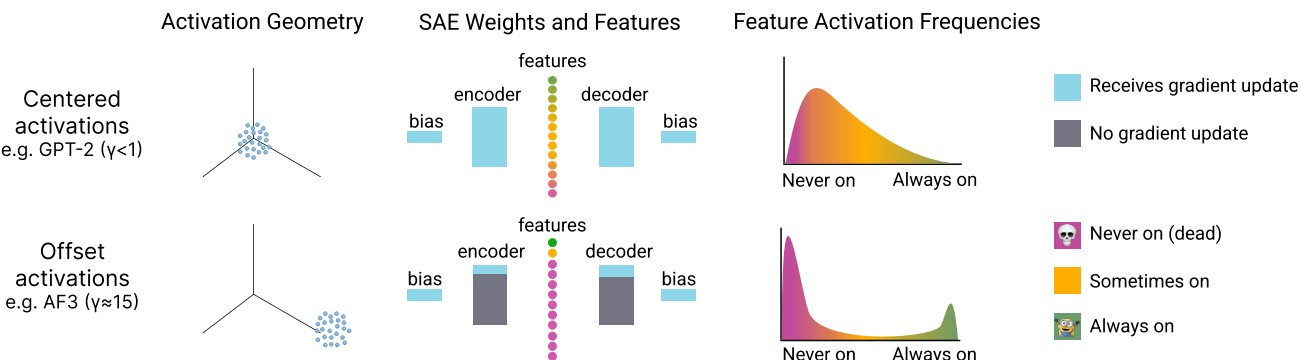

*Figure 1.* **Activation outliers predetermine feature fate at initialization.** (Left) Centered activations (e.g. GPT-2, $\gamma < 1$) versus activations with dimension-level outliers (e.g. AF3, $\gamma \approx 15$), where $\gamma = \|\boldsymbol{\mu}\|/\|\boldsymbol{\sigma}\|$ measures how much the activation mean dominates per-token variation. (Middle) In the offset case, only features positively aligned with the activation mean fire and receive gradient; anti-aligned features are dead from initialization. (Right) Centered features have input-dependent activation frequencies; features from offset activations are primarily always-on or never-on.

that responds to input content. When outlier dimensions inflate $\|\boldsymbol{\mu}\|$, the constant term dominates: features anti-aligned with the mean can never fire, features strongly aligned fire on everything, and only features roughly orthogonal to the mean remain input-dependent. The result is that most features have their fate set at initialization, not learned from data (Figure 1).

We formalize outlier severity as $\gamma = \|\boldsymbol{\mu}\|/\|\boldsymbol{\sigma}\|$, the ratio of the activation mean's magnitude to the per-token standard deviation magnitude. Features can die at initialization in two ways: by having permanently negative pre-activations (dead-by-ReLU) or by losing the top-$k$ competition (dead-by-TopK). From first principles, we derive that the initial dead-feature rate is a closed-form function of $\gamma$ alone, for both pathways. Synthetic experiments confirm this: increasing $\gamma$ monotonically increases death. On real activations from 454 model-layer combinations spanning language, vision, protein, and genomic models, $\gamma$ predicts initial death rates for both: Spearman $\rho = 0.89$ for dead-by-TopK, and $\rho = 0.82$ for dead-by-ReLU.

We characterize how dead features can revive during training. Revival has two pathways. Features that fail to rank in the top-$k$ on any input start ranking once competing features shrink during training. Features with permanently negative pre-activations recover only as the bias slowly absorbs the mean offset. This bias-learning pathway is the bottleneck: at high $\gamma$ the bias has further to travel, and dead features plateau at 75–90% even after 2M steps. AuxK (Gao et al., 2024) accelerates the first pathway but not the second, which explains why it helps at moderate $\gamma$ but not high $\gamma$.

Mean-centering validates this account directly. If the offset $\boldsymbol{\mu}$ causes death, removing it should eliminate outlier-induced death at initialization, and it does: mean-centering alone reduces dead features at initialization from 83% to near zero on ESM3 and from 98% to under 5% on Al-

phaFold3, without auxiliary losses in either condition. The surviving features are also higher quality. On ESM3 and DINOv3, two of the highest-death models in our suite, a mean-centered SAE produces more monosemantic features, and on ESM3 recovers more biological concepts than a baseline with four times the dictionary size. Mean-centering has been used in prior SAE work (Bricken et al., 2023b; Gao et al., 2024) but inconsistently and without a clear rationale for when it matters. Our analysis provides one: $\gamma$ identifies models where centering is necessary, and the recovery analysis explains why training alone cannot compensate in time.

Mean-centering does not eliminate all feature death. A few layers, primarily in protein and genomic models, retain residual death from a separate geometric cause: when activation variance is concentrated in a small number of directions, most features cannot win the top-$k$ competition regardless of the mean. PCA whitening equalizes variance across directions and eliminates this residual death (Appendix E).

**Contributions.**

1. **Outlier severity diagnostic.** We derive $\gamma = \|\boldsymbol{\mu}\|/\|\boldsymbol{\sigma}\|$ from first principles. Analytically, $\gamma$ determines the fraction of features with permanently negative pre-activations at initialization. The diagnostic matches synthetic experiments and correlates with death rates across 454 real model-layer combinations for both pathways (Spearman $\rho = 0.89$ and $\rho = 0.82$) with no fitting, letting practitioners assess SAE trainability before committing compute.

2. **Geometric cause of feature death.** We find that dead features in high-$\gamma$ models are a geometry problem, not a training dynamics problem. Outlier dimensions inflate the activation mean, creating input-independent pre-activation shifts that predetermine feature fate at initialization. This explains why the same SAE config-

urations produce near-zero dead features on GPT-2 but over 70% on AlphaFold3: the difference is not in the SAE but in the activation geometry.

3. **Recovery dynamics.** We characterize how dead features revive during training, finding two pathways with different timescales. Features that fail to rank in the top-$k$ on any input start ranking as competing features shrink. Features with permanently negative pre-activations recover only as the bias slowly absorbs the mean offset, a process that requires prohibitively long training at high $\gamma$. AuxK helps the first pathway but not the second.

4. **Validation via mean-centering.** Mean-centering eliminates outlier-induced death across all tested models, confirming the geometric account. It also provides a principled basis for when this preprocessing step is necessary (high $\gamma$) and why existing revival techniques cannot substitute for it.

## 2. Background

**Sparse autoencoders.** Neural networks appear to encode more concepts than they have dimensions, superimposing them as overlapping directions in activation space (Elhage et al., 2022). Sparse autoencoders (SAEs) try to undo this by treating activations as mixtures of unknown basis elements and solving for the basis (Olshausen & Field, 1996), using a learned encoder that maps activations to sparse codes in a single feedforward pass (Bricken et al., 2023b; Huben et al., 2024). Concretely, an SAE maps activations $\mathbf{x} \in \mathbb{R}^d$ to a higher-dimensional latent space $\mathbf{z} \in \mathbb{R}^n$ (with $n > d$), enforces sparsity on $\mathbf{z}$, and reconstructs $\mathbf{x}$:

$$\mathbf{z}_{\text{pre}} = \mathbf{W}_{\text{enc}}(\mathbf{x} - \mathbf{b}) + \mathbf{b}_{\text{enc}}$$
$$\mathbf{z} = \sigma(\mathbf{z}_{\text{pre}})$$
$$\hat{\mathbf{x}} = \mathbf{W}_{\text{dec}}^\top \mathbf{z} + \mathbf{b}$$

Here $\sigma$ is a sparsity-inducing nonlinearity and $\mathbf{b} \in \mathbb{R}^d$ is the *bias*, which centers the input before encoding and recenters the output after decoding. The decoder columns $\mathbf{W}_{\text{dec}}$ form the learned dictionary: each column is a direction in activation space, and the sparse code $\mathbf{z}$ indicates which directions are active for a given input.

The architectures differ mainly in how they enforce sparsity. ReLU SAEs use $\sigma = \text{ReLU}$ with an L1 penalty on $\mathbf{z}$. TopK SAEs apply ReLU and then keep only the $k$ largest positive pre-activations, zeroing the rest, so exactly $k$ features fire per input (Gao et al., 2024). JumpReLU SAEs learn per-feature thresholds, allowing adaptive sparsity (Rajamanoharan et al., 2024). We focus on TopK SAEs, but the core findings hold across architectures (Appendix D).

**Dead features and revival methods.** A dead feature is a dictionary element that never activates on any input. Be-

cause dead features receive zero gradient on their encoder weights, they cannot update toward useful directions: they are stuck. This can happen in two ways. First, any architecture that applies ReLU (or an equivalent threshold) will zero out a feature whose pre-activation is always negative, regardless of sparsity selection. Second, TopK SAEs introduce a competition-based pathway: a feature with positive pre-activations can still die if it never ranks among the top-$k$ on any input. Other architectures have analogous competition effects (JumpReLU features can be pushed below their learned thresholds), but TopK's hard selection makes this pathway especially stark.

Several methods try to revive dead features by injecting gradient signal where there would otherwise be none. AuxK (Gao et al., 2024) adds an auxiliary loss based on how well dead features reconstruct the residual error. Ghost gradients (Bricken et al., 2023b) backpropagate through a reconstruction computed from dead features. Resampling (Bricken et al., 2023b) periodically reinitializes dead feature weights toward directions with high reconstruction error. These methods treat dead features as a training dynamics problem: the features are stuck, so push them. Comparatively little attention has gone to why some models produce massive death rates in the first place; one recent exception, Wang et al. (2025), links death to low-rank structure in attention activations. Our focus is a different geometric cause: dimension-level activation outliers.

## 3. Dead Feature Rates Vary Dramatically Across Models

When we train SAEs on activations from a range of models and modalities, dead feature rates vary dramatically, from under 5% to over 95% (Figure 2a). SAEs were originally established as a tool for interpreting language models (Huben et al., 2024; Bricken et al., 2023b) but have since been applied across many domains including vision (Gorton, 2024; Joseph et al., 2025) and biology (Simon & Zou, 2025; Adams et al., 2025; Brixi et al., 2026). We train TopK SAEs with identical architecture, dictionary size, sparsity, and learning-rate sweep on source models spanning language, vision, protein, and genomic, using one SAE per model trained on a representative (mid-network) layer (full model list and hyperparameters in Appendix A). For all language models in our suite, we could train SAEs with no dead features; for many protein and vision models, even the best configuration left the majority dead. AuxK reduces death on some models but cannot bring the worst cases below 75%.

Among the models we examine, biology and vision models tend toward higher death than language models. But the variation within a single model can be just as large: ESM3 ranges from over 80% dead in early layers to under 20% in later layers, with the same SAE and same hyperparameters.

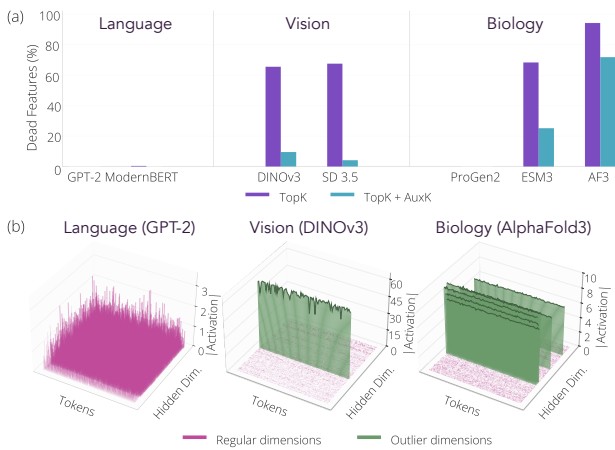

*Figure 2.* **Dead feature rates vary dramatically across models and coincide with dimension-level activation outliers.** (a) Dead feature rates for TopK SAEs across a subset of language, vision, protein, and genomic models (additional models in Figure S14). AuxK helps but does not resolve death where it is most severe. (b) Activation magnitude landscapes (LayerNorm-normalized per token) for three models. Green highlights dimensions with high mean and low per-token variance. High-death models are distinguished by dimensions whose magnitude is *consistently large* across tokens, not merely high-variance.

This implies the cause is something about each layer's activation distribution that some model families produce more often than others.

High-death layers share a specific activation pattern: certain dimensions take on large, near-constant activations across every token. This high-mean-low-variance signature differs from the per-token spikes typically studied as activation outliers (often measured by kurtosis; e.g., He et al., 2024). Lu et al. (2025) previously identified the same pattern in ESMFold. Compare GPT-2, where no dimension dominates, to ESM3 and AlphaFold3, where a handful of dimensions tower above the rest on every input (Figure 2b; per-model visualizations in Figure S1).

## 4. Why Outliers Cause Death at Initialization

One might expect outliers to cause dead features indirectly: distorting the loss, producing bad gradients, eventually breaking the encoder. But we observe something more direct: most features are already dead at initialization, before any gradients flow. The problem is not corrupted training dynamics, but rather a corrupted starting point.

### 4.1. Features anti-aligned with an outlier-inflated mean are initialized dead

Consider activations where one dimension has value $\sim 1000$ regardless of input, while other dimensions vary around zero with standard deviation $\sim 1$. At initialization the biases are zero, so the pre-activation for feature $i$ is $z_i = \mathbf{w}_i \cdot$

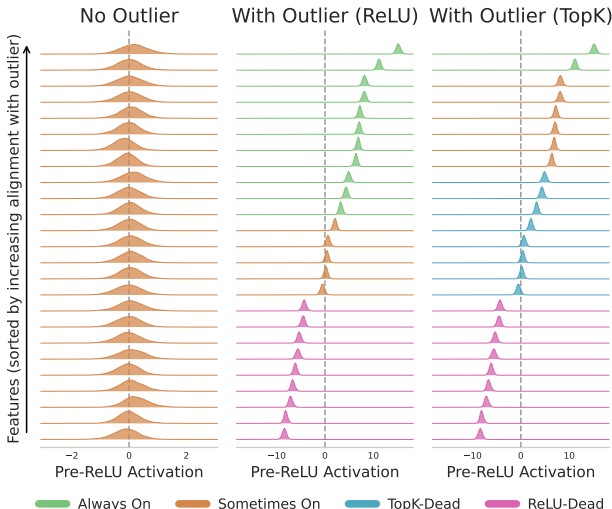

*Figure 3.* **A feature's alignment with the activation mean determines its fate at initialization.** Each row is one feature sorted by alignment with $\boldsymbol{\mu}$; ridgelines show its pre-activation distribution across inputs. Left (no outliers): all features are input-dependent. Middle (with outliers, ReLU SAE): anti-aligned features have permanently negative pre-activations (dead-by-ReLU); strongly aligned features fire on everything; only roughly orthogonal features remain input-dependent. Right (with outliers, TopK SAE): additional features with moderate positive pre-activations die because they never rank in the top $k$ (dead-by-TopK).

$\mathbf{x}$. If $w_{i,0} = +0.01$, dimension 0 alone contributes $+10$, swamping the varying dimensions. This feature fires on every input. If $w_{i,0} = -0.01$, the contribution is $-10$, and the feature never fires. Only features with $w_{i,0} \approx 0$ remain sensitive to input content.

This decomposition applies to arbitrary activation distributions, not just the single-outlier toy case. Decomposing the activation for an individual token as $\mathbf{x} = \boldsymbol{\mu} + (\mathbf{x} - \boldsymbol{\mu})$:

$$z_i = \underbrace{\mathbf{w}_i \cdot \boldsymbol{\mu}}_{\text{shift (constant)}} + \underbrace{\mathbf{w}_i \cdot (\mathbf{x} - \boldsymbol{\mu})}_{\text{signal (varies with input)}}$$

Whether a feature responds to input content or has its fate predetermined at initialization depends on which term dominates. The ratio $\gamma = \|\boldsymbol{\mu}\| / \|\boldsymbol{\sigma}\|$ quantifies this: high $\gamma$ means shifts dominate signals.

This produces two distinct failure modes (Figure 3). **Dead-by-ReLU:** features with large negative shifts ($\mathbf{w}_i \cdot \boldsymbol{\mu} \ll 0$) have pre-activations that no input can push above zero. **Dead-by-TopK:** features with moderate positive shifts pass ReLU but lose the TopK competition to features with large positive shifts; the same features win every selection. Both pathways have predictable extremes. Under symmetric initialization, roughly half of features align positively with outliers and half negatively, so dead-by-ReLU approaches 50% at high $\gamma$. For TopK, only the $k$ features most positively aligned with $\boldsymbol{\mu}$ ever win the competition, so dead-by-TopK approaches $1 - k/n$ (e.g. 99.2% at $k = 64$, $n = 8192$).

Beyond these extremes, we can derive analytically how $\gamma$ determines the death rate at any outlier severity.

## 4.2. Outlier Severity Predicts Death Rates Analytically

A feature dies when its negative shift exceeds the largest signal fluctuation across $N$ evaluation samples; as shift and signal are projections of random unit vectors onto fixed directions, we can use high-dimensional probability rules to analytically compute the probability of death-by-ReLU based on outlier severity (step by step derivation in Appendix B):

$$P(\text{dead-by-ReLU}) = \Phi\left(\frac{-C}{\gamma}\right)$$

where $\Phi$ is the standard normal CDF and $C = \Phi^{-1}(1 - 1/N)$ depends only on the number of evaluation samples ($C \approx 4.26$ for $N = 100{,}000$).

For dead-by-TopK the same setup applies, but the survival bar is higher: a feature has to land in the top $k$ pre-activations out of $n$ on at least one input. When $\gamma$ is large the bar is approximately fixed across inputs: the spread of shifts across features is roughly $\gamma$ times the spread of signals, so the top $k$ are essentially the $k$ features with the largest shifts no matter what the input is. The bar a feature must clear is then the $(1 - k/n)$ quantile of the shift distribution: the value such that only $k$ out of $n$ features have larger shifts. Substituting this for zero in the dead-by-ReLU derivation gives:

$$P(\text{dead-by-TopK}) \approx \Phi\left(t_k - \frac{C}{\gamma}\right), \quad t_k = \Phi^{-1}\left(1 - \frac{k}{n}\right)$$

The dead-by-ReLU formula is the $t_k = 0$ special case (full derivation in Appendix B.5).

## 4.3. Does $\gamma$ Predict Death in Practice?

**Synthetic experiments establish causality.** We generate activations with controlled $\gamma$ (single dominant outlier dimension; details in Appendix A.1.2) and train SAEs. Dead features increase monotonically with $\gamma$, with perfect rank correlation (Spearman $\rho = 1.0$). The theoretical curve $\Phi(-C/\gamma)$ matches observed dead-by-ReLU rates closely (Figure 4). Dead-by-ReLU plateaus near 50% at high $\gamma$; dead-by-TopK continues rising toward $1 - k/n$ as only the most positively-aligned features survive.

**Real activations show the same pattern.** Across 454 model-layer combinations spanning language, vision, protein, and genomic models, $\gamma$ predicts death rates with Spearman $\rho = 0.82$ for dead-by-ReLU and $\rho = 0.89$ for dead-by-TopK. These correlations are particularly notable given that the derivation assumes signal projections are Gaussian, which fails when activations are heavy-tailed (Appendix B.9), and that other geometric factors can also drive death (Appendix E). The correlation holds both across and

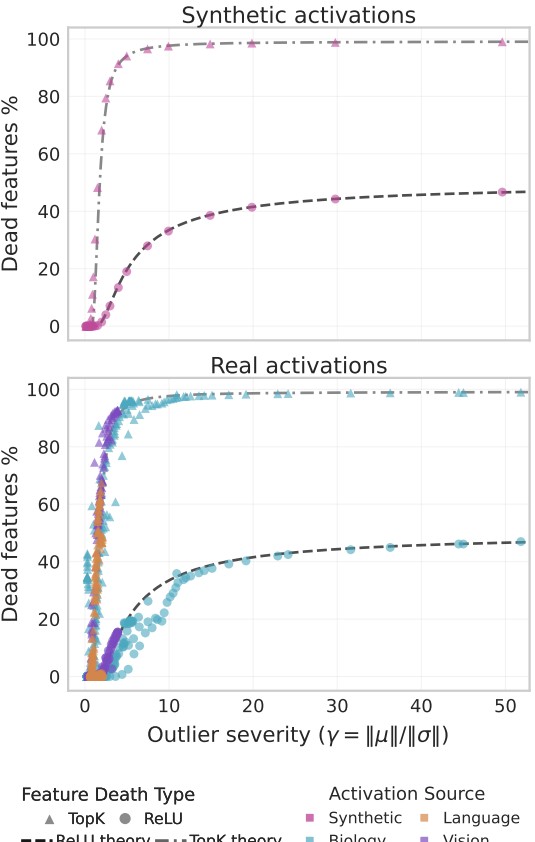

*Figure 4.* $\gamma$ **analytically predicts dead feature rates for both death pathways.** (a) Synthetic activations with controlled $\gamma$: dead features increase monotonically (Spearman $\rho = 1.0$). The ReLU theory curve $\Phi(-C/\gamma)$ matches dead-by-ReLU rates (circles); the TopK theory curve $\Phi(t_k - C/\gamma)$ tracks dead-by-TopK (triangles), becoming tight for $\gamma > 3$. The dotted line at 99.2% marks the $1 - k/n$ asymptote. (b) 454 real model-layer combinations spanning language, vision, protein, and genomic. The same relationships hold: Spearman $\rho = 0.82$ (dead-by-ReLU) and 0.89 (dead-by-TopK).

within model families (Figure 4). On real data we apply per-token LayerNorm (LN) (Ba et al., 2016) before computing $\gamma$. This preprocessing isolates the per-token outlier structure that drives feature death from between-token scale variation (largest in vision transformers), which would otherwise inflate $\|\boldsymbol{\sigma}\|$ uniformly across dimensions and cause raw $\gamma$ to understate outlier severity (Appendix B.8).

The formula slightly over-predicts dead-by-ReLU on some layers. The derivation treats signal projections as Gaussian, so across $N$ samples the largest signal concentrates near $C \approx 4.26$ standard deviations: a feature whose shift is more negative than this can never be pushed above zero. Real activations sometimes have heavier tails, and the largest sample then reaches well beyond $C$. Features with moderate negative shifts that the Gaussian formula declares dead are then occasionally rescued by these tail excursions. We diagnose

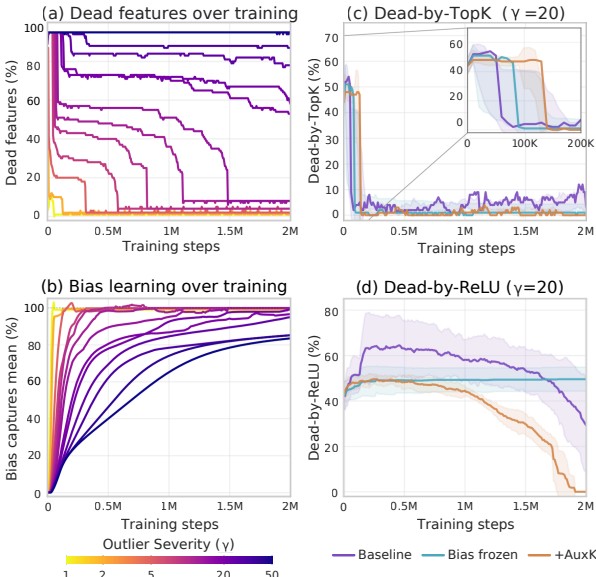

*Figure 5.* **Recovery from outlier-induced death is bottlenecked by slow bias learning.** All panels use synthetic activations with $\gamma$ set directly; the same two-phase pattern holds on real models (Figure S7). (a) Dead feature recovery slows as $\gamma$ increases. (b) Bias convergence to $\boldsymbol{\mu}$ slows as $\gamma$ increases. (c) Dead-by-TopK at $\gamma = 20$ drops steeply around 100K steps (inset) regardless of condition. (d) Dead-by-ReLU at $\gamma = 20$ is much slower: frozen bias prevents recovery entirely, and AuxK prevents the early spike but does not speed the underlying decline. Shaded bands: $\pm 1$ s.d. over 10 seeds.

this directly from per-dimension kurtosis of the activations themselves, which is cheap to compute and tracks where the formula breaks down (Appendix B.9, Figure S5).

# 5. Dead Features Revive During Training, but Recovery Is Slow

If outlier-induced death is just a mean offset, and SAEs have bias terms that can learn offsets, shouldn't training eventually fix itself? It does, but slowly. We study these dynamics with synthetic activations, which let us set $\gamma$ directly, and ablate individual components (bias, AuxK, sparsity competition) in isolation; the same dynamics hold on real models (Figure S7). Figure 5a tracks dead features over training across $\gamma$ values. At low $\gamma$, recovery finishes within a few hundred thousand steps. At high $\gamma$ ($\geq 30$), dead features plateau at 75–90% even after 2M steps. We find that the two death pathways (dead-by-ReLU and dead-by-TopK) recover through different mechanisms, one fast and one slow, and the slow one is bottlenecked by the bias learning the activation mean, which takes longer for larger means.

## 5.1. The two death pathways are resolved by different parameters

Decomposing dead features at $\gamma = 20$ by death pathway (Figure 5c,d) reveals two very different dynamics. Dead-by-TopK features (Figure 5c) revive within $\sim$200K steps in all three conditions: the baseline, an ablation where the bias parameter is held frozen during training, and +AuxK. This produces the sharp early drop visible in Figure 5a. Dead-by-ReLU features (Figure 5d) decline much more slowly: the baseline continues to drop across the remaining 1.8M steps but does not fully reach zero by the end of training, and under frozen bias they remain permanently dead (individual feature trajectories in Appendix C.2).

To understand why the two pathways have such different revival timescales, we look at how each one actually revives. Dead features in either pathway receive no gradient on their own encoder weights, since TopK or ReLU zeros their decoder contribution and no signal flows back. Revival therefore requires other parameters to change. For dead-by-ReLU features, the only parameter that can shift a stuck-negative pre-activation above zero is the bias, so these features revive only as the bias updates. For dead-by-TopK features, the dominant mechanism is encoder weights of alive features changing: as those alive features (which do receive gradient) reduce their own activations during training, dead-by-TopK features rise into the top-$k$. The bias could in principle also reshuffle the TopK rankings, but empirically does not drive revival. Freezing the bias during training makes the asymmetry visible: TopK revival proceeds almost unchanged, but dead-by-ReLU features remain permanently dead.

A side effect of the TopK revival mechanism shows up in Figure 5d: dead-by-ReLU rates *increase* during the first $\sim$200K steps, the same window where TopK revival is happening. When alive features reduce their activations to free up TopK slots, some get pushed below zero in the process and become dead-by-ReLU. This collateral death increases with $\gamma$ (Appendix C.1), and explains why the sharp early drop in Figure 5a becomes less visible at high $\gamma$: TopK revival is partially offset by new ReLU deaths, so total dead count stays relatively flat despite active turnover underneath. This collateral death phenomenon will also help us understand AuxK's effect on dead-by-ReLU recovery in Section 5.3; see Appendix C for further analysis of these recovery dynamics.

## 5.2. Bias learning is the bottleneck

Why does dead-by-ReLU recovery scale so much worse with $\gamma$ than dead-by-TopK? It turns out bias learning is the bottleneck, and it slows with $\gamma$. Figure 5b shows bias convergence to the mean over training (one curve per $\gamma$): at $\gamma \leq 5$, the bias reaches $\sim$99% of $\boldsymbol{\mu}$ within $\sim$200K steps;

at $\gamma \approx 20$, only $\sim 90\%$ after 2M steps; at $\gamma \geq 30$, only 50–70%.

Features, by contrast, capture $\boldsymbol{\mu}$ much faster than the bias. Feature weights are multiplied by inputs, so small weight changes have effects that scale with input magnitude; the bias is added directly, with effects independent of input scale. As $\|\boldsymbol{\mu}\|$ grows, features can keep up via small weight changes while the bias takes proportionally longer. Once alive features capture $\boldsymbol{\mu}$, they reduce the reconstruction residual that drives the bias gradient, slowing bias learning further (Appendix C.3).

### 5.3. AuxK prevents collateral death but does not speed up bias learning

Comparing Figure 5c and Figure 5d, we see AuxK's most visible benefit is on dead-by-ReLU (Figure 5d), not dead-by-TopK (Figure 5c). TopK revival proceeds on a similar timescale regardless of whether AuxK is used. This is surprising. AuxK's auxiliary loss applies ReLU before computing reconstructions from dead features, so features with negative pre-activations receive no gradient from it. Given that we observe bias learning as the bottleneck for dead-by-ReLU recovery, a natural guess is that AuxK works by speeding up bias learning. But that's not what happens.

AuxK does not noticeably speed up bias convergence (Appendix C.3); rather, it reduces collateral death. Recall from Section 5.1 that TopK revival creates new dead-by-ReLU features: once features aligned with the mean learn to reconstruct it, they start decreasing their activations; the threshold drops, opening TopK slots but also pushing some active features below zero. AuxK provides gradient to dead-by-TopK features, helping them stabilize above zero rather than crossing into dead-by-ReLU. The reduction in dead-by-ReLU that AuxK achieves comes from **preventing collateral death-by-ReLU**, not from recovering features that were dead-by-ReLU from the start.

This provides one explanation for why AuxK's benefit depends on $\gamma$. At moderate $\gamma$ (10–20), collateral death accounts for a large share of persistent dead-by-ReLU features, so preventing it is the difference between full recovery and permanent death. At high $\gamma$ ($\geq 30$), most dead-by-ReLU features were dead from initialization, born anti-aligned with $\boldsymbol{\mu}$, not created during threshold collapse. Preventing collateral death addresses a shrinking fraction of the total problem, so AuxK provides diminishing benefit (Appendix C.1).

The common thread across all of these dynamics is $\boldsymbol{\mu}$: it predetermines feature fate at initialization, creates both death pathways, and imposes a recovery timescale that grows with its magnitude. Rather than waiting for the bias to slowly learn $\boldsymbol{\mu}$ during training, we can initialize it with the mean

directly.

## 6. Mean-centering eliminates outlier-induced death and improves feature quality

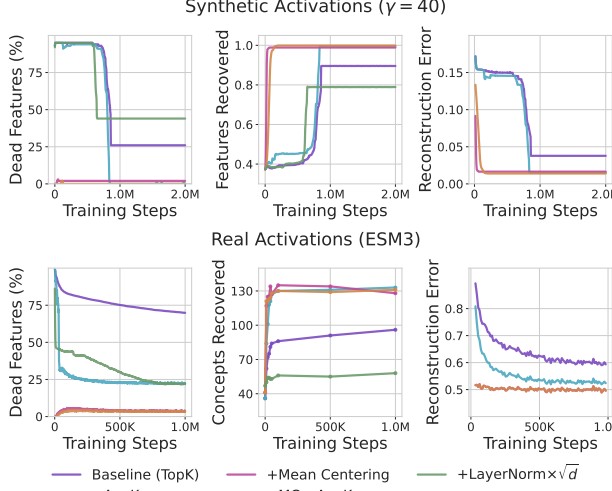

*Figure 6.* **Mean-centering sidesteps the recovery bottleneck: features are never dead to begin with.** Top: synthetic data ($\gamma = 40$); mean-centering achieves near-zero death, near-perfect recovery of ground-truth features, and lower reconstruction error. Bottom: ESM3 layer 24 ($\gamma \approx 8$); baseline reaches around 75% dead by end of training, AuxK plateaus $\sim 25\%$, mean-centering stays near zero and recovers more biological concepts than baseline.

When we initialize $\mathbf{b}$ with $\boldsymbol{\mu}$ (effectively mean-centering our activations), the pre-activation becomes:

$$z_i = \mathbf{w}_i \cdot (\mathbf{x} - \boldsymbol{\mu}) + b_{\text{enc}}$$

The shift term $\mathbf{w}_i \cdot \boldsymbol{\mu}$ vanishes and all features have pre-activations centered around zero, varying only with input content. No features are born dead from this mean offset.

Rather than subtracting $\boldsymbol{\mu}$ at runtime, we fold it into bias initialization: set the bias to the activation center (we use the geometric median by default, which works better than the arithmetic mean on certain models; details and per-model comparison in Appendix D.5). This is mathematically equivalent to runtime centering but requires no additional runtime computation.

### 6.1. Mean-centering eliminates dead features at initialization

If our explanation of outlier-induced death is correct, mean-centering should eliminate it from the start. We test this on synthetic data with controlled $\gamma = 40$, and on our representative middle layer from ESM3 (L24, $\gamma \approx 8$).

On synthetic data (Figure 6 top), mean-centering achieves near-zero dead features throughout training and reduces reconstruction error by an order of magnitude; the base-

line never recovers within 2M steps (∼8B tokens), and AuxK reaches similar dead-feature counts only via threshold collapse around 800K steps (the same dynamics from Section 5, accelerated because AuxK provides gradient to otherwise-stuck features). On ESM3 layer 24 (Figure 6 bottom), baseline dead features reach around 75% by end of training; AuxK drops to ∼25% but plateaus; mean-centering starts and stays near zero. LayerNorm with $\sqrt{d}$ rescaling (Templeton et al., 2024) reduces ESM3 death to ∼20% but recovers fewer concepts than baseline, and does not match mean-centering on synthetic data.

Figure S14 summarizes results across all source models, each trained on its representative middle layer for 1M steps: mean-centering eliminates outlier-induced death consistently, with improvements proportional to $\gamma$. Low-$\gamma$ models (GPT-2, Pythia) show minimal change; high-$\gamma$ models (ESM3, AlphaFold3) show dramatic reductions.

Mean-centering also reduces sensitivity to learning-rate choice: baseline dead-feature rates vary widely across the LR sweep, while mean-centered rates stay consistently low. While we focus on TopK SAEs (where we can easily fix sparsities across models), ReLU and JumpReLU SAEs also benefit from mean-centering (Appendix D).

## 6.2. Surviving features are more useful

Beyond reducing dead-feature counts, mean-centering produces features that better recover the structure we expect SAEs to find. Two settings let us compare SAE features to *expected* ones: synthetic data with ground-truth feature directions we constructed, and protein models where features should encode known biological structure. In both, mean-centered SAEs recover more of the expected features.

On synthetic data, mean-centered SAE features align with the ground-truth feature directions much more closely than baseline. The mean maximum cosine similarity (MMCS) between learned and ground-truth features is 0.97 for mean-centered SAEs vs. 0.38 for baseline (full metric definition in Appendix A.3).

On our representative middle layer of ESM3, mean-centered SAEs capture substantially more known biological concepts (Table 1; methods in Appendix A.3). Mean-centering outperforms a 4× larger dictionary here: a mean-centered SAE with 2048 features captures more concepts (100) than a baseline SAE at 8192 (73), at a fraction of the compute.

Surviving features are also more monosemantic. We computed Monosemanticity Scores (Pach et al., 2025) on high-death models from two domains: DINOv3 (vision, with CLIP-ViT as the independent embedding model) and ESM3 (protein, with ESM2). The metric measures semantic coherence of each feature's top-activating inputs. On ESM3, mean-centered SAEs produce more monosemantic features

| Dict size | Baseline | +Mean Center |
|---|---|---|
| 2048 | 61 | 100 |
| 4096 | 69 | 121 |
| 8192 | 73 | 127 |

*Table 1.* **Mean-centering outperforms a 4× larger dictionary.** Number of biological concepts (out of 187 SwissProt concepts) captured by at least one SAE feature with $F_1 > 0.7$, on ESM3 representative middle layer, $k = 16$. A mean-centered 2048-feature SAE recovers more concepts than a baseline 8192-feature SAE.

across the score distribution (e.g., 2× more features above MS>0.5 at the same dictionary size; Figure S16), while raw neurons score near zero (confirming SAEs perform real decomposition). On DINOv3, baseline features have almost no meaningful monosemanticity while mean-centered features hold nonzero scores across most of the dictionary. Across our highest-death settings (synthetic activations, protein models, and vision models), all three evaluations show mean-centered features are higher quality.

## 6.3. When mean-centering isn't enough

Mean-centering addresses outlier-induced death, but this does not account for all death: other geometric causes appear in some layers, and features can still die over the course of training depending on hyperparameters.

Mean-centering eliminates outlier-induced death at the representative layer for every model in our suite except one. The exception is Evo1, where 73% of features remain dead at initialization and ∼58% after training. Why does the fix that works everywhere else fail here? Evo1's activations are extremely low-rank: just 4 of 4096 principal components capture 99% of the variance. When variance concentrates in so few directions, only the features aligned with those few PCs can win the TopK competition, and the rest never fire. Centering removes the mean offset but leaves this directional imbalance intact. The same low-rank pattern, less severe, appears in the early layers (L1–L2) of DINOv3-7B, ESM3, ESM2-3B, and ProGen2-base; their middle layers (our typical SAE training sites) are unaffected, which is why mean-centering suffices there. Mean-centering always helps, but its *sufficiency* depends on whether the post-centering covariance is rich enough for diverse features to compete.

PCA whitening modifies the activations to equalize variance across principal components, cleanly eliminating Evo1's residual death and reaching 0% on every other affected layer. Active Subspace Initialization (Wang et al., 2025) instead initializes features in the data's high-variance directions without modifying the activations; on our pathologically low-rank cases it requires more aggressive settings than Wang et al. tested, leaving its effectiveness less clear. Together, $\gamma$ and an effective-rank measurement are enough

to choose preprocessing per layer: MC when $\gamma$ is high, PCA or ASI when the post-centering rank is low. See Appendix E for full analysis of the low-rank death mechanism, the per-layer diagnostic, and the comparison of fixes.

Additionally, large learning rates or sparsity penalties can kill features during training, even from a death-free initialization (Appendix D).

## 7. Related Work

**Activation preprocessing for SAE training.** Mean-centering has appeared in prior SAE work, but its usage has been inconsistent across the literature, with no clear account of when it matters or why. Bricken et al. (2023b) initialized the bias to the activation mean (equivalent to centering), but later Anthropic releases omit this step (Lindsey et al., 2024). Gao et al. (2024) use centering, noting only that it "helps training" without ablation. Conerly et al. (2025) initialize encoder biases so each feature fires at a target rate, a related technique that implicitly accounts for the mean. $\gamma$ predicts when centering is necessary, and the recovery analysis (Section 5) explains why training alone cannot compensate in time.

Separately, Saraswatula & Klindt (2025) show that PCA whitening improves SAE feature quality on SAEBench (Karvonen et al., 2025) by making the optimization landscape more convex. We use whitening for a complementary reason: to revive dead features in low-rank layers.

**Origins of dimension-level outliers.** The outliers we study are an instance of a well-documented phenomenon: large transformers develop a small number of hidden dimensions with activation magnitudes far exceeding the rest (Dettmers et al., 2022). Their causes have been traced to optimizer and architectural choices (Elhage et al., 2023; He et al., 2024), and other work proposes architectural mitigations (Bondarenko et al., 2023; Hu et al., 2024; Luo et al., 2025). These works address a different type of outlier (per-token spikes, captured by their kurtosis and infinity norm metrics) and propose changes applied during pretraining rather than handling outliers in existing models. Recent work has found that both dimension level outlier activations and attention sinks function as important rescaling factors which stabilize training (Qiu et al., 2026).

## 8. Discussion

Feature death in SAEs is often not a training problem at all. In models with high outlier severity ($\gamma$), activation geometry determines which features will live and die before the first gradient step. The two death pathways (dead-by-ReLU and dead-by-TopK) originate from the same geometric cause but recover on very different timescales. Bias learning is the

bottleneck at high $\gamma$, explaining why AuxK rescues some models but not others.

We did not initially expect the mechanism to be this direct. A more natural hypothesis (and the one we started with) was that outliers corrupt gradients or distort the loss landscape, causing training to slowly break down. Instead, the damage is done at initialization, and no amount of training recovers it within practical compute budgets.

Reviving dead features yields better SAEs at the same size: mean-centered SAEs match a $4\times$ larger baseline on concept recovery and are more monosemantic (Figure S16).

**Practical guidance.** For most models, initialize the SAE bias to the geometric median of activations: this eliminates outlier-induced death at zero runtime cost. The exception is models with intrinsically low-rank activations, where a few principal components carry nearly all the variance. The post-LN effective rank of the activation covariance, computable on a single batch, predicts when this occurs: below $\sim 2\%$ of hidden dimension, MC alone leaves substantial residual death and PCA whitening reaches 0% in every case (Table S8). Active Subspace Initialization (Wang et al., 2025) is an alternative that modifies the encoder initialization rather than the activations; it works well in moderately low-rank settings but has unclear effectiveness in the more extreme cases we see.

Variable death rates also affect cross-layer methods that chain feature dictionaries across layers, such as transcoders (Dunefsky et al., 2024); mean-centering stabilizes effective capacity, which may help.

**Limitations.** Our analysis focuses on TopK SAEs and residual stream activations; ReLU and JumpReLU SAEs also exhibit outlier-induced death but experience additional training-time death from sparsity pressure that we don't analyze. $\gamma$ explains much of the variation in dead-feature rates at initialization, but other geometric properties also contribute: heavy tails reduce death below the formula's prediction (Appendix B.9), and low-rank structure causes additional death beyond what $\gamma$ captures (Appendix E).

**Open questions.** Why outlier severity differs across architectures and domains is an open practical question. A similar open thread is LayerNorm, another common SAE preprocessing step typically motivated by easier hyperparameter transfer (Templeton et al., 2024). We find that LN changes training dynamics but neither resolves outlier-induced death nor matches mean-centering on feature recovery (Figure 6, Appendix D). How LN affects dynamics across more models, particularly those with large per-token scale variation, is worth further study.

## Acknowledgments

We thank Mert Yuksekgonul for helpful feedback on the first draft of this paper, and members of the Zou lab for valuable conversations. E.S. is supported by NSF GRFP (grant no. DGE-2146755). E.A. is supported by NIH T32 training grant (grant no. 1T32GM158494-01).

## Impact Statement

This paper presents work whose goal is to advance the field of Machine Learning, specifically mechanistic interpretability. By improving our ability to understand the internal representations of neural networks across domains (language, vision, biology), this work may contribute to AI safety and alignment research, and improve our ability to study diverse model types including frontier biological models which can provide scientific insights through analysis.

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

## Appendix Contents

# A. Experimental Setup

## A.1. Datasets

### A.1.1. REAL-WORLD ACTIVATION DATASETS

We extract activations from a range of pretrained models spanning language, vision, protein, and genomic modalities. Per-model details are in Table S1.

| Model Name | Rep. Layer | # Layers Swept | Dataset |
|---|---|---|---|
| GPT-2 (Radford et al., 2019) | 6 | 11 | OpenWebText (Gokaslan et al., 2019) |
| Pythia-410M (Biderman et al., 2023) | 12 | 23 | OpenWebText |
| Pythia-70M (Biderman et al., 2023) | — | 5 | OpenWebText |
| ModernBERT-Large (Warner et al., 2025) | 14 | 27 | OpenWebText |
| ModernBERT (Warner et al., 2025) | — | 21 | OpenWebText |
| DINOv2-L (Oquab et al., 2024) | 12 | 23 | CIFAR-10 (Krizhevsky et al., 2009) |
| DINOv2-B (Oquab et al., 2024) | — | 11 | CIFAR-10 |
| DINOv3-7B (Siméoni et al., 2025) | 20 | 39 | CIFAR-10 |
| DINOv3-B (Siméoni et al., 2025) | 6 | 11 | CIFAR-10 |
| Stable Diffusion 3.5 Large (Esser et al., 2024) | 37 | 37 | CIFAR-10 |
| ESM2-3B (Lin et al., 2023) | 18 | 35 | Swiss-Prot (UniProt-Consortium, 2025) |
| ESM2-650M (Lin et al., 2023) | 16 | 32 | Swiss-Prot |
| ESM2-35M (Lin et al., 2023) | — | 11 | Swiss-Prot |
| ESM3 (Hayes et al., 2025) | 24 | 47 | Swiss-Prot |
| ProGen2-large (Nijkamp et al., 2023) | 15 | 31 | Swiss-Prot |
| ProGen2-base (Nijkamp et al., 2023) | — | 26 | Swiss-Prot |
| gLM2 (Cornman et al., 2024) | 15 | 32 | Swiss-Prot |
| AlphaFold3 (Abramson et al., 2024) | Pairformer single rep (no recycle) | 1 | UniRef (Suzek et al., 2015) |
| Evo1 (Nguyen et al., 2024) | 14 | 31 | EMBL European Nucleotide Archive (Yuan et a |
| | **Total** | **454** | |

*Table S1.* Models, datasets, and per-model layer counts. *Rep. Layer* gives the representative middle-of-network layer used in main figures (e.g. Figures 2, 6, S14); a dash indicates the model is included only in the cross-model $\gamma$-vs-death analysis (Figure 4) and has no rep-layer experiments. *# Layers Swept* gives the number of layers each model contributes to the cross-model sweep, excluding the embedding layer (layer 0). All extractions use 10M tokens.

### A.1.2. SYNTHETIC ACTIVATION DATASETS

We generate synthetic data following a sparse linear model. Each sample $\mathbf{x} \in \mathbb{R}^{50}$ is generated as:

$$\mathbf{x} = \mathbf{F}^\top \mathbf{c} + \boldsymbol{\mu}_{\text{outlier}} \tag{1}$$

where $\mathbf{F} \in \mathbb{R}^{100 \times 50}$ contains 100 ground truth feature directions (random unit vectors), $\mathbf{c} \in \mathbb{R}^{100}$ is a sparse activation vector, and $\boldsymbol{\mu}_{\text{outlier}} \in \mathbb{R}^{50}$ is a constant bias vector.

For each sample, exactly 5 features are active (5% sparsity), selected uniformly at random. Active coefficients are drawn from $c_i \sim |\mathcal{N}(0,1)| + 0.5$ (shifted half-normal), ensuring positive activations with minimum 0.5.

The outlier bias vector has a single non-zero entry, $(\boldsymbol{\mu}_{\text{outlier}})_0$, set to a configurable value while the remaining entries are zero. Varying this value produces synthetic data spanning approximately $\gamma \in [1, 50]$, where $\gamma = \|\boldsymbol{\mu}\|/\|\boldsymbol{\sigma}\|$ as defined in the main text; the specific $\gamma$ values used are visible as data points in Figure 4.

By default, each synthetic SAE uses dictionary size 100 (matching the number of ground truth features), TopK with $k = 5$ (matching the number of active features per sample), batch size 4096, and trains for 1–2M steps on 1M samples (specific durations are listed with figures). These defaults are varied in ablations in Appendix B.6.

## A.2. SAE Architectures and Training

We use ReLU SAEs as described in (Lindsey et al., 2024), TopK SAEs as described in (Gao et al., 2024), and Jump-ReLU SAEs as described in (Rajamanoharan et al., 2024).

For SAEs trained on real-world activations, unless otherwise noted we use a batch size of 4096 tokens, dictionary size 8192, and 100,000 training steps ($\approx$410M tokens per training run). We sweep over learning rates in the range $10^{-5}$ to

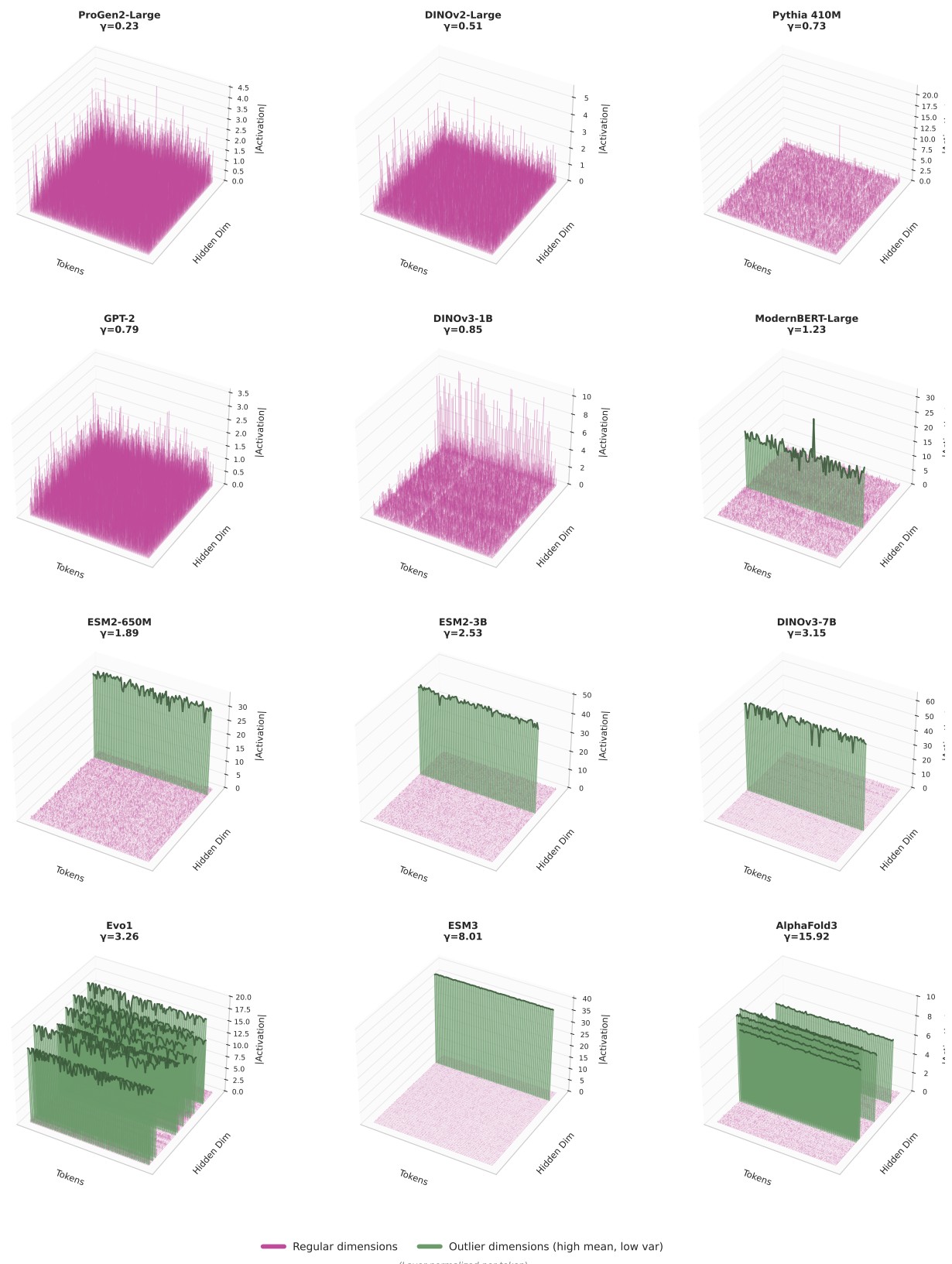

*Figure S1.* **Dimension-level outliers appear as persistent high-magnitude ridges across tokens.** Activation magnitude for each model at its representative layer, ordered by increasing $\gamma$. Green: dimensions with high mean and low per-token variance. LayerNorm-normalized per token.

| Parameter | Default value |
|---|---|
| Data dim ($d$) | 50 |
| Ground truth features ($m$) | 100 |
| Sparsity | 0.05 (5 active per sample) |
| Dataset size | 1,000,000 samples |
| Dictionary size | 100 |
| TopK $k$ | 5 |
| Batch size | 4096 |
| Training steps | 1–2,000,000 |

*Table S2.* Synthetic data parameters.

$10^{-3}$ and select the run with the lowest MSE. For SAEs trained on synthetic activations, training configuration is given in Appendix A.1.2 (Table S2).

TopK SAE (Gao et al., 2024) hyperparameters:

| Hyperparameter | Default | Description |
|---|---|---|
| TopK ($k$) | 64 | Number of active features selected per example |
| AuxK coefficient | 1/32 | Weight on the auxiliary TopK loss term (AuxK) |
| Examples until dead | 256,000 | Number of examples with no activation after which a unit is considered "dead" |

*Table S3.* TopK hyperparameters.

JumpReLU SAE (Rajamanoharan et al., 2024) hyperparameters:

| Hyperparameter | Value | Description |
|---|---|---|
| $\theta$ | 0.001 | Initial JumpReLU threshold value |
| $\epsilon$ | 0.001 | Straight-through estimator (STE) bandwidth |
| L0 penalty weight | 1 | L0 sparsity penalty weight (see Eq. 16 in Rajamanoharan et al. (2024)) |
| target L0 | 64 | Targeted sparsity |

*Table S4.* JumpReLU hyperparameters.

ReLU SAE (Huben et al., 2024) hyperparameters:

| Hyperparameter | Values | Description |
|---|---|---|
| L1 coefficient | $\{10^{-4}, 10^{-3}, 10^{-2}, 10^{-1}\}$ | Weight on the L1 sparsity penalty |
| Ghost gradients | $\{$True, False$\}$ | Whether to use ghost gradients to provide learning signal for features that are inactive (helps revive/learn rarely-active or dead features). |

*Table S5.* ReLU hyperparameters.

## A.3. Evaluation Metrics

We evaluate SAEs on the following metrics, applied as appropriate to each experimental setting:

- **Dead feature percentage**: Fraction of latents that never activate on 100k held-out examples. During training, we consider features dead if they have not activated for 256k examples.

- **MSE**: Mean squared reconstruction error $\|\mathbf{x} - \hat{\mathbf{x}}\|^2$.

- **MMCS (synthetic only)**: Mean max cosine similarity with ground-truth features. For each ground-truth feature direction, we compute its maximum cosine similarity with any learned dictionary direction, then average these maxima across ground-truth features. MMCS is high when every ground-truth feature has at least one learned feature pointing in (approximately) the same direction.

- **Biological concept coverage**: For each of 187 SwissProt biological concepts (UniProt-Consortium, 2025), we train a linear probe per SAE feature predicting that concept on 10,000 random SwissProt sequences, following Simon &

Zou (2025). We evaluate at ESM3 layer 24. We report the number of concepts that have at least one SAE feature achieving $F_1 > 0.7$, as a coverage metric for biologically meaningful structure. To avoid evaluating concepts unlikely to have associated features, we selected 187 concepts (out of a total of 833 concepts) that had previously shown at least moderate performance in ESM2, defined as an F1 score greater than 0.3.

- **Monosemanticity Score (MS)**: Defined by Pach et al. (2025) as the semantic coherence of each feature's top-activating inputs under an independent embedding model. We use CLIP-ViT for DINOv3 features and ESM2 for ESM3 features. Higher MS indicates a feature's top activations cluster tightly in the independent model's embedding space, suggesting the feature represents a single meaningful concept.

The 100k held-out threshold for the dead-feature metric is empirically justified: dead-feature rates plateau by $\sim 10^5$ examples across modalities (Figure S2).

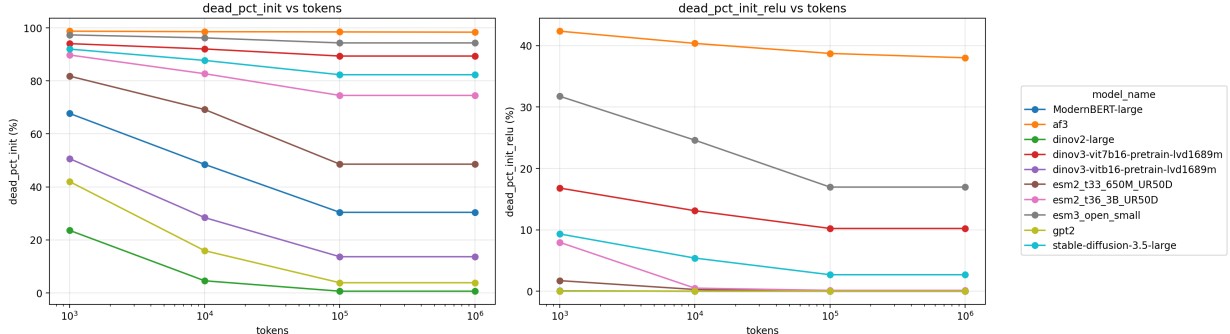

*Figure S2.* **Dead feature rates plateau around 100,000 evaluation examples.** Left: total dead features; right: dead-by-ReLU. Both stabilize by $10^5$ examples across modalities.

## B. Why $\gamma$ predicts dead features: derivation and validation

We want to predict, at random initialization, how many SAE features die. A feature dies when its pre-activation never crosses some threshold across the evaluation set: zero for ReLU, the input-dependent rank-$k$ boundary for TopK. We show below that both reduce, to a good approximation, to a closed-form expression in a single scalar quantity: $\gamma = \|\boldsymbol{\mu}\|/\|\boldsymbol{\sigma}\|$, the activation mean's norm divided by the per-dimension standard deviation's norm. The dead-by-ReLU rate is $\Phi(-C/\gamma)$ and the dead-by-TopK rate is $\Phi(t_k - C/\gamma)$ (Section 4.2).

**Approach.** The derivation has four steps:

1. Each feature's encoder weight $\mathbf{w}_i$ is a random unit vector. Its projection onto any fixed vector is approximately Gaussian (Appendix B.1).

2. Use this to decompose the pre-activation $z_i(\mathbf{x}) = s_i + r_i(\mathbf{x})$ into a data-independent *shift* $s_i = \mathbf{w}_i \cdot \boldsymbol{\mu}$ and a data-dependent *signal* $r_i(\mathbf{x}) = \mathbf{w}_i \cdot (\mathbf{x} - \boldsymbol{\mu})$, both with Gaussian distributions parameterized by $\boldsymbol{\mu}$ and $\boldsymbol{\sigma}$.

3. For each death pathway (dead-by-ReLU and dead-by-TopK), find the shift threshold below which no sample can rescue the feature.

4. Compute the fraction of features whose shift falls below this threshold; the dead-by-TopK extension follows the same approach with a different threshold.

**Assumptions.** The key assumption: for a fixed encoder weight $\mathbf{w}_i$, the signal $\mathbf{w}_i \cdot (\mathbf{x} - \boldsymbol{\mu})$ is approximately Gaussian over the data distribution. This holds when activations are not too heavy-tailed; heavier-tailed activations rescue features the formula would predict dead (Appendix B.9 quantifies when). We also assume evaluation samples are approximately independent.

**Measurement note.** When computing $\gamma$ on real data, we apply per-token LayerNorm before measuring $\boldsymbol{\mu}$ and $\boldsymbol{\sigma}$. This removes between-token scale variation that would otherwise inflate $\|\boldsymbol{\sigma}\|$ and understate the outlier severity (Appendix B.8).

### B.1. Projections of Random Unit Vectors Are Approximately Gaussian

Encoder weights are initialized by sampling i.i.d. entries from $\mathcal{N}(0, 1)$ and normalizing each row to unit norm, which is equivalent to sampling uniformly from the unit sphere in $d$ dimensions.

The projection of such a random unit vector $\mathbf{w}$ onto any fixed vector $\boldsymbol{\mu}$ is approximately Gaussian with mean zero and variance $\|\boldsymbol{\mu}\|^2/d$. Write $\mathbf{w} = \mathbf{g}/\|\mathbf{g}\|$ where $\mathbf{g} \sim \mathcal{N}(0, \mathbf{I}_d)$. The unnormalized projection $\mathbf{g} \cdot \boldsymbol{\mu}$ is exactly $\mathcal{N}(0, \|\boldsymbol{\mu}\|^2)$, since it is a weighted sum of independent Gaussians. In high dimensions, $\|\mathbf{g}\|$ concentrates around $\sqrt{d}$, so dividing by $\|\mathbf{g}\|$ scales the variance by $1/d$:

$$\mathbf{w} \cdot \boldsymbol{\mu} \ \sim \ \mathcal{N}\left(0, \ \frac{\|\boldsymbol{\mu}\|^2}{d}\right) \tag{2}$$

### B.2. Pre-activations Decompose into a Fixed Shift and a Varying Signal

The pre-activation for feature $i$ on input $\mathbf{x}$ decomposes as:

$$z_i(\mathbf{x}) = \mathbf{w}_i \cdot \mathbf{x} = \underbrace{\mathbf{w}_i \cdot \boldsymbol{\mu}}_{s_i \text{ (shift)}} + \underbrace{\mathbf{w}_i \cdot (\mathbf{x} - \boldsymbol{\mu})}_{r_i(\mathbf{x}) \text{ (signal)}} \tag{3}$$

The shift $s_i$ is locked in once encoder weights are drawn; it is identical for every input the SAE will ever process. The signal $r_i(\mathbf{x})$ is the part that responds to what the input contains. If the shift dominates, the feature fires (or doesn't) regardless of input.

**Shift distribution.** The shift $s_i = \mathbf{w}_i \cdot \boldsymbol{\mu}$ is a projection of a random unit vector onto the fixed mean $\boldsymbol{\mu}$. Applying the result from the previous subsection:

$$s_i \sim \mathcal{N}\left(0, \ \frac{\|\boldsymbol{\mu}\|^2}{d}\right) \tag{4}$$

**Signal distribution.** The signal $r_i(\mathbf{x}) = \mathbf{w}_i \cdot (\mathbf{x} - \boldsymbol{\mu})$ varies with each input. For a fixed input, the same projection argument gives variance $\|\mathbf{x} - \boldsymbol{\mu}\|^2/d$, which differs across inputs. To characterize typical signal magnitude:

$$\mathbb{E}_{\mathbf{x}}\big[\|\mathbf{x} - \boldsymbol{\mu}\|^2\big] = \sum_{j=1}^{d} \mathrm{Var}(x_j) = \sum_{j=1}^{d} \sigma_j^2 = \|\boldsymbol{\sigma}\|^2 \tag{5}$$

When $d$ is large and dimensions are not strongly correlated, this sum of $d$ squared deviations concentrates tightly around its expectation, with relative fluctuation shrinking as $1/\sqrt{d}$. So for a typical input:

$$r_i(\mathbf{x}) \ \sim \ \mathcal{N}\left(0, \ \frac{\|\boldsymbol{\sigma}\|^2}{d}\right) \quad \text{(for typical inputs)} \tag{6}$$

### B.3. Features Die When Their Shift Exceeds $C$ Standard Deviations

A feature is dead-by-ReLU if $z_i(\mathbf{x}) = s_i + r_i(\mathbf{x}) < 0$ for all $N$ evaluation samples. Each sample is one chance for the signal to rescue the feature: survival requires just a single input where the signal overcomes the negative shift. The question is how negative the shift must be for none of $N$ samples to provide enough signal.

Let $\sigma_r = \|\boldsymbol{\sigma}\|/\sqrt{d}$ denote the typical signal standard deviation. For a feature with shift $s_i = -t$ (where $t > 0$), the probability that a single sample's signal exceeds $t$ is $P(r_i(\mathbf{x}) > t) = 1 - \Phi(t/\sigma_r)$. Across $N$ independent samples, the expected number that exceed $t$ is $N \cdot [1 - \Phi(t/\sigma_r)]$.

We define the lethal threshold as the shift where this expected count equals 1, below which fewer than one sample is expected to rescue the feature: The exact cutoff matters less than it might seem: $C$ varies only from 3.7 to 4.8 as $N$ ranges from $10^4$

to $10^6$, so predicted death rates are insensitive to the precise choice. Setting the expected count to 1 and solving:

$$N \cdot \left[ 1 - \Phi\left(\frac{t}{\sigma_r}\right) \right] = 1 \tag{7}$$

$$\Phi\left(\frac{t}{\sigma_r}\right) = 1 - \frac{1}{N} \tag{8}$$

$$\frac{t}{\sigma_r} = \Phi^{-1}\left(1 - \frac{1}{N}\right) \equiv C \tag{9}$$

The lethal threshold is therefore $t^* = C \cdot \sigma_r = C \cdot \|\boldsymbol{\sigma}\|/\sqrt{d}$. Features with shift more negative than $-t^*$ would need a signal fluctuation so large that we expect fewer than one such event across all $N$ samples.

**Values of $C$:**

| $N$ | $C = \Phi^{-1}(1 - 1/N)$ |
|---|---|
| $10^3$ | 3.09 |
| $10^4$ | 3.72 |
| $10^5$ | 4.26 |
| $10^6$ | 4.75 |

For our evaluation with $N = 100{,}000$: $C = 4.26$.

### B.4. The Death Rate Follows From $\gamma$ in Closed Form

All that remains is to ask what fraction of features drew a shift this unlucky. Since $s_i \sim \mathcal{N}(0, \|\boldsymbol{\mu}\|^2/d)$, the probability that a feature's shift falls below $-t^* = -C \cdot \|\boldsymbol{\sigma}\|/\sqrt{d}$ is:

$$P(\text{dead-by-ReLU}) = P\left( s_i < -C \cdot \frac{\|\boldsymbol{\sigma}\|}{\sqrt{d}} \right) \tag{10}$$

$$= P\left( \frac{s_i}{\|\boldsymbol{\mu}\|/\sqrt{d}} < \frac{-C \cdot \|\boldsymbol{\sigma}\|/\sqrt{d}}{\|\boldsymbol{\mu}\|/\sqrt{d}} \right) \tag{11}$$

$$= P\left( Z < \frac{-C \cdot \|\boldsymbol{\sigma}\|}{\|\boldsymbol{\mu}\|} \right) \quad \text{where } Z \sim \mathcal{N}(0, 1) \tag{12}$$

$$= \Phi\left( \frac{-C}{\gamma} \right) \tag{13}$$

where $\gamma = \|\boldsymbol{\mu}\|/\|\boldsymbol{\sigma}\|$. The embedding dimension $d$ drops out entirely: both the shift variance ($\|\boldsymbol{\mu}\|^2/d$) and the signal variance ($\|\boldsymbol{\sigma}\|^2/d$) scale as $1/d$, so their ratio is dimension-free. This is why $\gamma$ works as a diagnostic across models with embedding dimensions ranging from 384 (AlphaFold3) to 4096 (DINOv3, Evo1).

The final result:

$$\boxed{P(\text{dead-by-ReLU}) = \Phi\left( \frac{-C}{\gamma} \right), \quad C = \Phi^{-1}(1 - 1/N)} \tag{14}$$

**Limiting behavior.** As $\gamma \to 0$ (no outliers), the argument of $\Phi$ goes to $-\infty$ and $P(\text{dead}) \to 0$: without outliers, random features are not born dead. As $\gamma \to \infty$ (extreme outliers), the argument approaches 0 and $P(\text{dead}) \to 0.5$: feature fate is determined entirely by the sign of $\mathbf{w}_i \cdot \boldsymbol{\mu}$, with positive and negative alignment equally likely under symmetric initialization.

**Multiple outlier dimensions.** The formula does not assume outliers are concentrated in a single dimension. The shift $s_i = \mathbf{w}_i \cdot \boldsymbol{\mu}$ and its variance $\|\boldsymbol{\mu}\|^2/d$ depend only on the norm of $\boldsymbol{\mu}$, not on how it is distributed across coordinates. For example, one dimension with a mean of 1000 and ten dimensions each with a mean of 316 produce the same $\gamma$ and the same predicted death rate.

### B.5. Extension to Dead-by-TopK

The dead-by-ReLU derivation asks whether a feature's shift is so negative that no input can push its pre-activation above zero. Dead-by-TopK raises the bar: a feature must not merely be positive, but rank among the top $k$ out of $n$ features on at least one input.

**The threshold is approximately fixed across inputs at high $\gamma$.** At high $\gamma$, the shift $s_i = \mathbf{w}_i \cdot \boldsymbol{\mu}$ dominates every feature's pre-activation: shifts are spread across features by a factor $\gamma$ more than the signal $r_i(\mathbf{x})$ can move any single feature. The signal therefore cannot reorder features whose shifts already differ by several times its range, and the top-$k$ selection picks essentially the same $k$ features on every input: the $k$ with the largest shifts. The bar a feature must clear is therefore approximately constant across inputs: it is the $(k+1)$-th largest shift across features, equivalently the $(1 - k/n)$ quantile of the shift distribution.

Since shifts are i.i.d. draws from $\mathcal{N}(0, \|\boldsymbol{\mu}\|^2/d)$, this quantile is:

$$\tau_k \;=\; \Phi^{-1}\left(1 - \frac{k}{n}\right) \cdot \frac{\|\boldsymbol{\mu}\|}{\sqrt{d}} \;=\; t_k \cdot \frac{\|\boldsymbol{\mu}\|}{\sqrt{d}}, \quad t_k \equiv \Phi^{-1}\left(1 - \frac{k}{n}\right) \tag{15}$$

For $k = 64$ and $n = 8192$: $t_k = \Phi^{-1}(0.9922) = 2.42$.

**Death criterion.** A feature is dead-by-TopK if its pre-activation never exceeds $\tau_k$ on any of $N$ evaluation samples:

$$\max_{j=1,\ldots,N} \left[s_i + r_i(\mathbf{x}_j)\right] \;<\; \tau_k \tag{16}$$

Following the same argument as Appendix B.3, the feature survives only if some sample's signal exceeds the gap $\tau_k - s_i$. The expected number of rescuing samples is $N \cdot [1 - \Phi((\tau_k - s_i)/\sigma_r)]$, and setting this to 1 gives the lethal threshold: a feature dies when

$$s_i \;<\; \tau_k \;-\; C \cdot \sigma_r \;=\; t_k \cdot \frac{\|\boldsymbol{\mu}\|}{\sqrt{d}} \;-\; C \cdot \frac{\|\boldsymbol{\sigma}\|}{\sqrt{d}} \tag{17}$$

**Death rate.** The fraction of features whose shift falls below this threshold is:

$$P(\text{dead-by-TopK}) = P\left(s_i < t_k \cdot \frac{\|\boldsymbol{\mu}\|}{\sqrt{d}} - C \cdot \frac{\|\boldsymbol{\sigma}\|}{\sqrt{d}}\right) \tag{18}$$

$$= \Phi\left(\frac{t_k \cdot \|\boldsymbol{\mu}\|/\sqrt{d} - C \cdot \|\boldsymbol{\sigma}\|/\sqrt{d}}{\|\boldsymbol{\mu}\|/\sqrt{d}}\right) \tag{19}$$

$$= \Phi\left(t_k - \frac{C}{\gamma}\right) \tag{20}$$

where $\gamma = \|\boldsymbol{\mu}\|/\|\boldsymbol{\sigma}\|$ as before. The dead-by-ReLU formula $\Phi(-C/\gamma)$ is the special case $t_k = 0$: the ReLU threshold is zero, so only the signal term matters.

**Limiting behavior.** As $\gamma \to \infty$, the argument of $\Phi$ approaches $t_k = \Phi^{-1}(1 - k/n)$, giving $P(\text{dead}) \to 1 - k/n$. For $k = 64$, $n = 8192$, this is 99.2%: only the $k$ features most positively aligned with $\boldsymbol{\mu}$ survive, and the rest are dead regardless of input content.

### B.6. $\gamma$ predicts death across SAE width and sparsity $k$

The main-text analysis fixes the SAE width and sparsity for clarity. To check that $\gamma$ predicts death across these axes too, we trained TopK SAEs on synthetic activations spanning a range of $\gamma$ values, jointly sweeping SAE width $\in \{10\times, 25\times, 50\times, 100\times\}$ the input dimension and sparsity $k \in \{5, 16, 32, 64, 100\}$. Figure S3 shows the result.

Effect of SAE Width on Dead Features at Initialization (d=100)

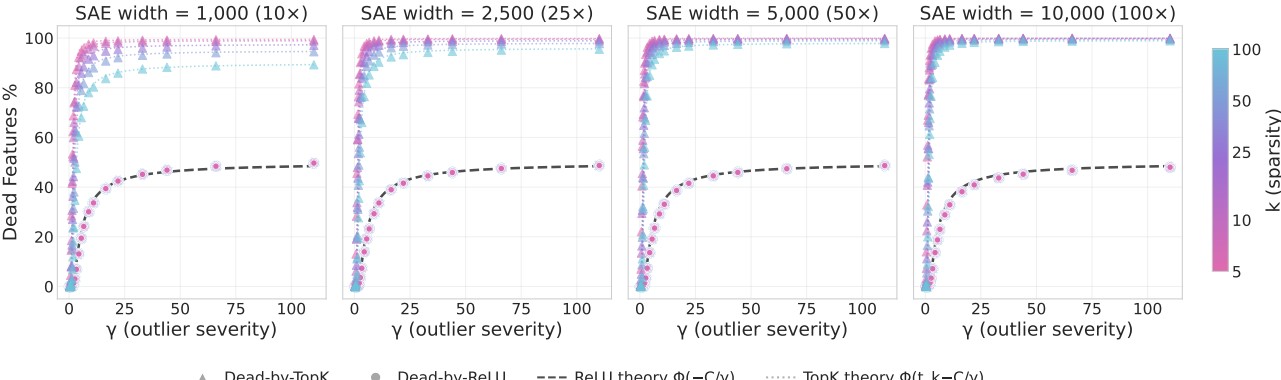

*Figure S3.* $\gamma$ **predicts death rates across SAE width and sparsity.** Each panel shows a different SAE width ($10\times$ to $100\times$ the input dimension); within each panel, color encodes sparsity $k \in \{5, 16, 32, 64, 100\}$ and marker shape distinguishes the two death pathways (triangles: dead-by-TopK; circles: dead-by-ReLU). Death rates rise monotonically with $\gamma$ on every (width, $k$) combination, confirming that the $\gamma$ diagnostic is robust to SAE hyperparameter choice. Dotted lines: per-$k$ dead-by-TopK theory $\Phi(t_k - C/\gamma)$. Dashed line: dead-by-ReLU theory $\Phi(-C/\gamma)$.

## B.7. Death tracks $\gamma$ whether driven by $\mu$ or $\sigma$

The $\gamma$ formula combines $\|\mu\|$ and $\|\sigma\|$ into a single ratio. To test that the ratio (rather than either component alone) is the operative quantity, we ran a complementary sweep to Figure S3: instead of varying $\gamma$ through the outlier mean, we fix $\mu$ and sweep the per-dimension std $\sigma$. Death rates collapse onto the same $\gamma$-determined curves (Figure S4), confirming that the ratio $\gamma$ itself, not $\mu$ or $\sigma$ alone, is what predicts death.

Varying σ (fixed μ=50): SAE Width vs Dead Features at Initialization (d=100)

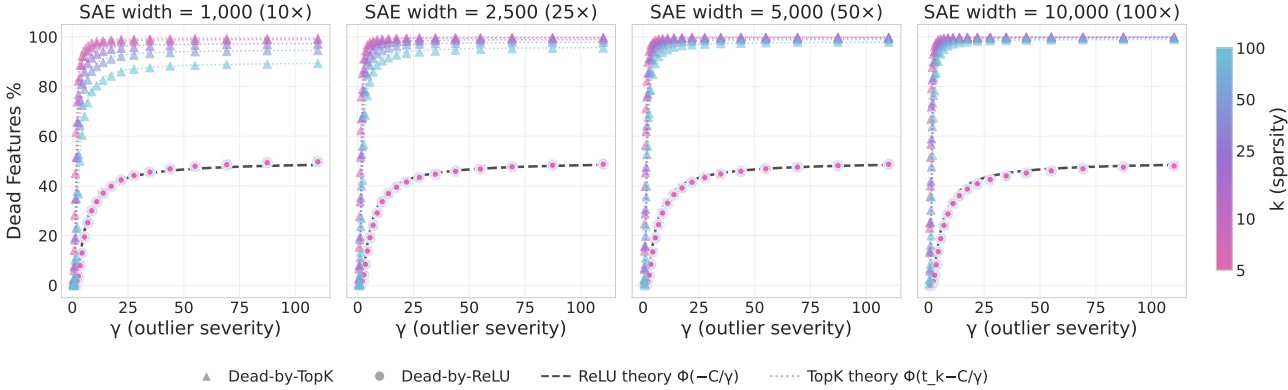

*Figure S4.* **Death tracks $\gamma$ whether driven by $\mu$ or $\sigma$.** Same four-panel sweep over SAE width and sparsity $k$ as Figure S3, but $\gamma$ is now varied by sweeping $\sigma$ at fixed $\mu = 50$ rather than by sweeping the outlier mean. Death rates collapse onto the same $\gamma$-determined curves as in Figure S3, confirming that $\gamma = \|\mu\|/\|\sigma\|$ is the operative quantity, not $\mu$ or $\sigma$ alone. Markers and theory lines as in Figure S3.

## B.8. Per-token scale variation affects $\gamma$ measurement

$\gamma$ is computed from $\mu$ and $\sigma$ measured across tokens, but the structure that drives feature death is per-token: a feature is alive iff it ranks among the top $k$ pre-activations on at least one token. These rankings on any single token are unaffected by that token's overall norm, since rescaling a token multiplies every feature's pre-activation by the same positive factor and leaves their order unchanged. So feature death does not depend on how token norms vary across the dataset. Raw $\gamma$ does: tokens with large norms inflate $\sigma$ more than $\mu$ (variance squares deviations). When tokens have very different norms (typical of vision transformers), raw $\gamma$ understates outlier severity without the actual death rate changing.

To measure the per-token directional structure that actually drives death, we normalize each token to unit norm (per-token LayerNorm) before computing $\mu$ and $\sigma$. SAEs are trained on the original unnormalized activations; we use the normalized activations only for the $\gamma$ measurement. Per-token rescaling to a target norm (e.g., $\sqrt{d_{\text{model}}}$) (Templeton et al., 2024) has the same effect.

The empirical effect is exactly what we'd expect: per-token LN changes $\gamma$ for models with high scale variation and leaves it largely unchanged for models with naturally uniform token norms (Table S6). Vision transformers exhibit the largest per-token scale variation: DINOv3-7B layers have $\gamma_{\text{LN}}/\gamma_{\text{raw}}$ ratios up to $16.9\times$ (median $5.3\times$). Without normalization, DINOv3-7B's within-model correlation between $\gamma$ and death is *inverted* ($\rho = -0.61$), because the layers with the worst outliers also have the most per-token scale variation, masking the signal. Language models show moderate effects (GPT-2: median $2.5\times$). Protein language models are barely affected (ESM3: median $1.1\times$), because their tokens already have similar norms.

| Model | Domain | $n_{\text{layers}}$ | $\gamma_{\text{LN}}/\gamma_{\text{raw}}$ |
|---|---|---|---|
| DINOv3-7B | Vision | 40 | $5.3\times$ (1.1–16.9) |
| ModernBERT-Large | Language | 28 | $4.3\times$ (1.0–10.7) |
| GPT-2 | Language | 12 | $2.5\times$ (1.0–4.3) |
| ESM3 | Protein | 48 | $1.1\times$ (1.0–1.3) |
| AF3 | Protein | 1 | $1.1\times$ |

*Table S6.* **Per-token LayerNorm changes the measured $\gamma$.** $\gamma_{\text{LN}}/\gamma_{\text{raw}}$: median ratio across layers (range in parentheses). Per-token normalization to a target norm (e.g., $\sqrt{d_{\text{model}}}$) (Templeton et al., 2024) has the same effect.

## B.9. Where the Formula Is Less Accurate: Heavy Tails Rescue Dead Features

In Figure 4 (bottom panel), most dead-by-ReLU points (circles) track the theoretical curve closely, but some layers consistently fall below it: most visibly in ESM3 and ESM2-650M, where layers at $\gamma \approx 5$–15 have 5–15% fewer dead-by-ReLU features than predicted. The cause is heavy-tailed signals. The derivation assumes the signal $r_i(\mathbf{x})$ is Gaussian, in which case its maximum across $N = 100{,}000$ samples is approximately $C \approx 4.26$ standard deviations of the signal distribution (i.e., $C\sigma_r$, where $\sigma_r$ is the per-feature signal std). A feature with shift $s_i$ below $-C\sigma_r$ cannot be rescued by any sample and is dead. When the signal is actually heavy-tailed, the maximum across $N$ samples can reach 8 or 9 $\sigma_r$; features with shifts down to $-8\sigma_r$ or $-9\sigma_r$ are then alive in reality but predicted dead by the formula.

Figure S5 illustrates this for ESM3. For a single random feature direction, the signal distribution on layer 8 ($\kappa \approx 0$) matches the Gaussian density and the observed maximum is $4.5\sigma_r$, close to the predicted $4.3\sigma_r$ (panel a, middle). On layer 29 ($\kappa = 1.11$), the tails are visibly heavier and the maximum reaches $8.3\sigma_r$ (panel a, right). This is not a fluke of one direction: across 5,000 random feature directions, the median maximum signal on layer 29 is $8.0\sigma_r$, nearly double the $4.3\sigma_r$ median for Gaussian data (panel b). A feature with shift $s_i = -6\sigma_r$ would be permanently dead under Gaussian signals but alive on layer 29, because several samples exceed $+6\sigma_r$.

What makes some layers heavy-tailed? Per-dimension kurtosis of the activation matrix $X$ predicts the kurtosis of random projections $r$ at $\rho = 0.97$ (panel c). This is useful because per-dimension kurtosis is cheap to compute directly from activations, while signal kurtosis would require sampling many random projections; the tight correlation lets us diagnose formula breakdown from a single pass over $X$. Per-dimension kurtosis also predicts which layers fall below the curve in Figure 4: it correlates with formula overprediction at $\rho = -0.78$ for ESM3 and $\rho = -0.62$ for ESM2-650M (panels d–e). Layers with near-zero kurtosis match the formula to within 1%; layers with kurtosis above 0.5 show 5–9% overprediction. The effect is concentrated in middle layers of ESM3 (layers 17–31, where $\kappa$ peaks at $\sim 1.4$); early and late layers are near-Gaussian and match the formula closely (panel f). ESM2-650M has elevated kurtosis only at its earliest layers ($\kappa > 3$ at layers 0–1), likely because the embedding layer's output is a token-vocabulary lookup, which produces a non-Gaussian distribution of activations across tokens.

The formula $\Phi(-C/\gamma)$ is therefore a useful upper bound on dead-by-ReLU rates and a close approximation whenever excess kurtosis is below $\sim 0.3$, which covers most model-layer combinations in our dataset.

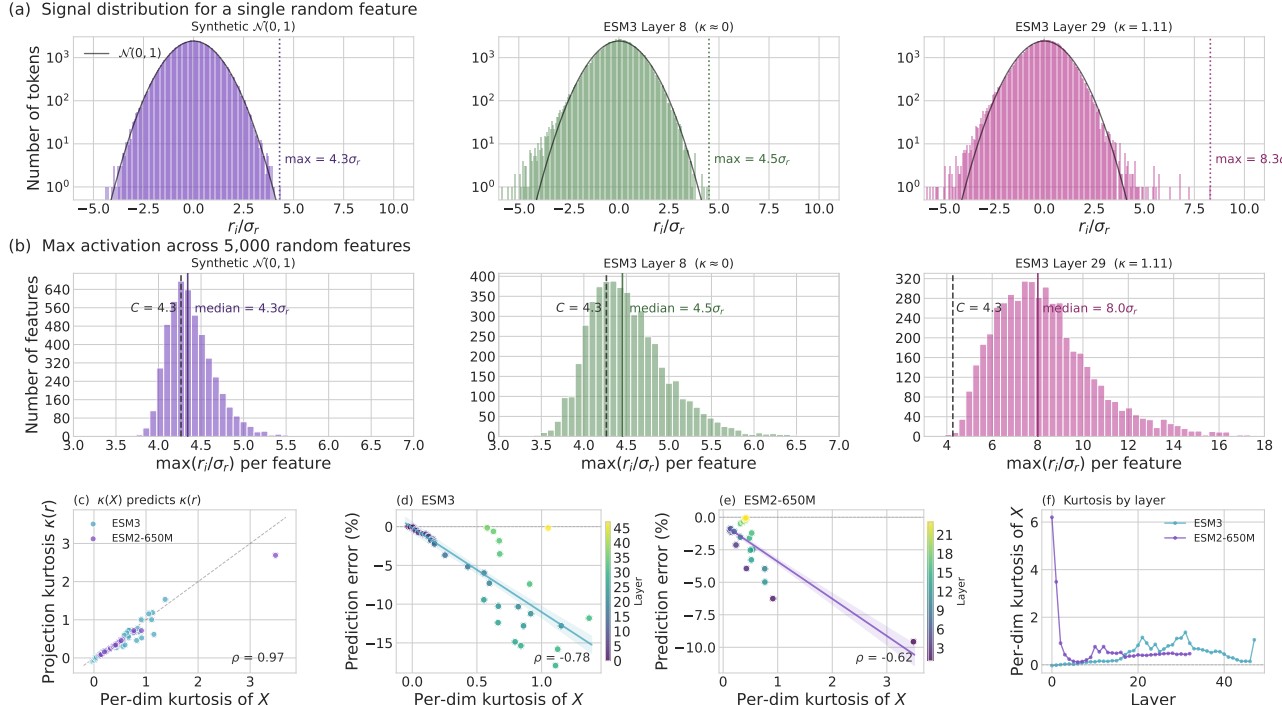

*Figure S5.* Heavy-tailed embedding dimensions break the Gaussian assumption underlying the death formula. **(a)** Signal distribution $r_i/\sigma_r$ for a single random feature direction, comparing synthetic Gaussian data, ESM3 layer 8 (near-Gaussian, $\kappa \approx 0$), and ESM3 layer 29 (heavy-tailed, $\kappa = 1.11$). The black curve shows the expected $\mathcal{N}(0,1)$ density; dotted lines mark the observed maximum. Layer 29's tails extend far beyond the Gaussian prediction. **(b)** Distribution of $\max(r_i/\sigma_r)$ across 5,000 random feature directions. The dashed line marks the Gaussian-predicted maximum $C = 4.3$; the solid line marks the empirical median. For Gaussian and layer 8, the median is near $C$; for layer 29, the median is $8.0\sigma_r$, nearly double, rescuing features the formula predicts should be dead. **(c)** Per-dimension kurtosis of $X$ predicts the kurtosis of random projections ($\rho = 0.97$), confirming the mechanistic link. **(d–e)** Higher kurtosis correlates with larger formula overprediction (ESM3: $\rho = -0.78$; ESM2-650M: $\rho = -0.62$), colored by layer depth. **(f)** Per-dimension kurtosis varies by layer, peaking in middle layers of ESM3 and early layers of ESM2-650M.

## C. Extended Analysis of Feature Revival Dynamics

Section 5 traces the recovery dynamics at $\gamma = 20$ through two stages: a fast TopK revival driven by TopK threshold collapse, followed by a slow dead-by-ReLU revival driven by bias learning. The subsections below extend that picture across $\gamma$, at the single-feature level, and through the bias-learning bottleneck. Synthetic experiments use the SAE setup described in Appendix A.

### C.1. Collateral death increases with $\gamma$

To see how the two-stage structure varies with $\gamma$, we sweep $\gamma \in \{2, 8, 14, 20, 30, 50\}$ and track dead-by-TopK and dead-by-ReLU trajectories through the collapse window (Figure S6; collapse windows highlighted in orange). Each panel reports the change in dead-by-TopK ($\Delta$TopK), the change in dead-by-ReLU ($\Delta$ReLU), and their ratio during the collapse window.

At $\gamma = 2$, the two-stage structure is barely visible. Dead-by-TopK drops from $\sim$25 to near zero within the collapse window. Dead-by-ReLU barely changes (ratio 0.04): almost no features cross below zero during threshold collapse. At $\gamma = 8$, TopK revival is larger ($\Delta$TopK $= -61\%$) but collateral death remains small (ratio $-0.03$): the threshold drop opens slots without pushing many features below zero.

The picture changes at moderate $\gamma$. At $\gamma = 14$, dead-by-ReLU rises by 15% during the collapse window while dead-by-TopK drops by 52% (ratio 0.29). At $\gamma = 20$, the ratio reaches 0.50: half of the TopK revival is offset by new ReLU deaths. This is why total dead count in Figure 5a shows a smaller net drop at $\gamma = 20$ than one might expect from the TopK revival alone.

At $\gamma \geq 30$, collateral death roughly equals TopK revival (ratio $\sim$1.0). Dead-by-ReLU rises by 31% at $\gamma = 30$ while TopK drops by 32%. Total dead count appears flat not because nothing is happening, but because TopK revival and collateral

ReLU death are in approximate balance.

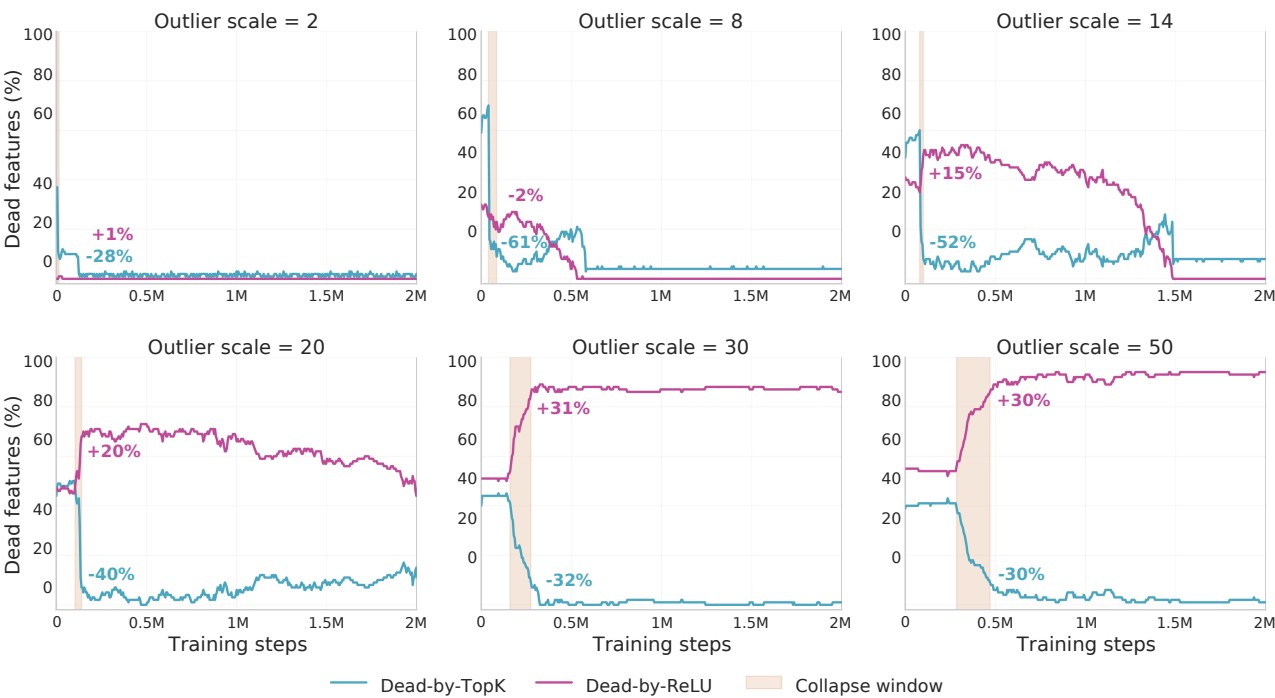

*Figure S6.* **Collateral death increases with $\gamma$.** Dead-by-TopK (cyan) and dead-by-ReLU (pink) over training at six $\gamma$ values. Orange bands mark the collapse window (period of steepest TopK decline). Inset text reports the change in each pathway during the collapse window and their ratio. At low $\gamma$, TopK revival proceeds with minimal collateral death. At $\gamma \geq 30$, the ratio approaches 1.0: new ReLU deaths roughly offset TopK revival, explaining why total dead count in Figure 5a appears flat at high $\gamma$ despite active turnover.

The same dynamics appear on real models. Figure S7 shows the dead-by-TopK / dead-by-ReLU decomposition over training for three high-$\gamma$ models: AlphaFold3, Stable Diffusion 3.5 layer 37, and ESM3 layer 30. All three exhibit the same pattern: fast TopK revival during the collapse window, simultaneous conversion of features to dead-by-ReLU, then a plateau. Collateral death is substantial in every case; on SD3.5, dead-by-ReLU rises by 76 percentage points during collapse, nearly matching the 79-point TopK drop. The bottom row shows the bottleneck directly: after 100K steps, the bias has captured less than 0.3% of the activation mean in every model. These real activations have much larger absolute scales than our synthetic experiments at similar $\gamma$, so the required bias shift is correspondingly large. Recovery time scales with the absolute $\|\boldsymbol{\mu}\|$ (Appendix C.3), and bias learning is too slow to make meaningful progress within practical training budgets. Until the bias catches up, dead-by-ReLU features have no mechanism to revive, which is why they plateau.

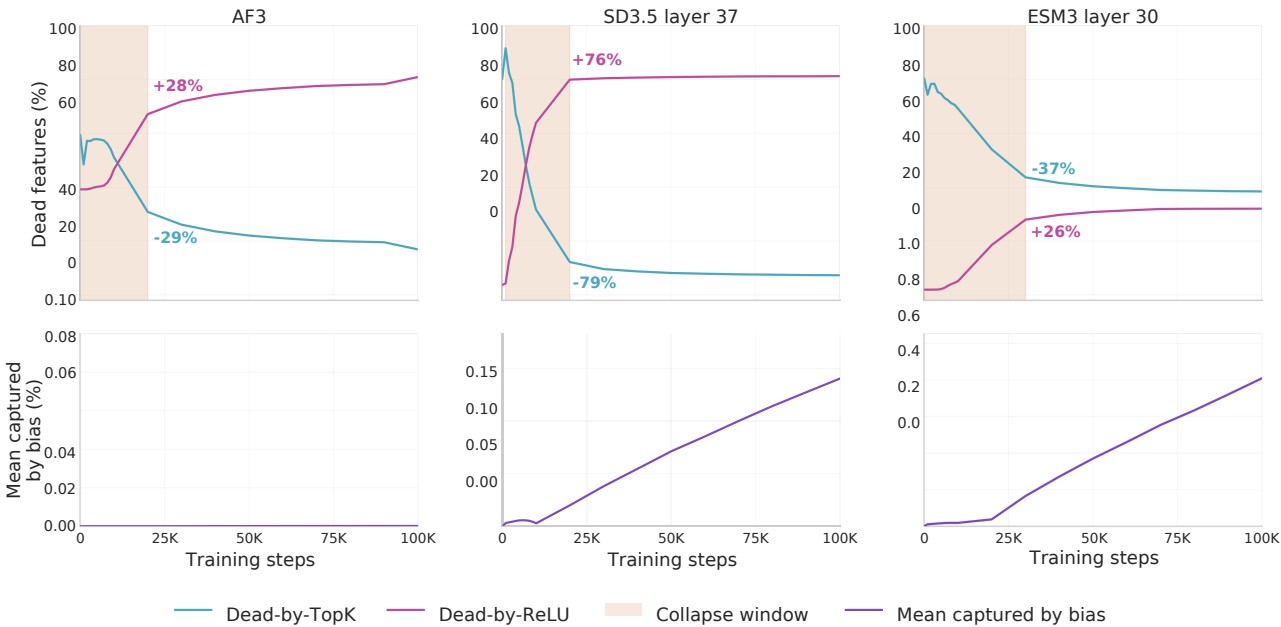

*Figure S7.* **The two-phase recovery dynamics from Figure 5 hold on real models.** Dead-by-TopK (teal) and dead-by-ReLU (magenta) over training for three real models: AlphaFold3, Stable Diffusion 3.5 layer 37, and ESM3 layer 30. Shaded bands mark the collapse window. Top row: all three exhibit the same pattern seen in synthetic experiments: fast TopK revival during collapse (teal drops), simultaneous conversion of features to dead-by-ReLU (magenta rises), then a plateau. Annotated percentages show the change in each pathway during the collapse window. Bottom row: bias has captured less than 0.3% of the activation mean after 100K steps in every case. These models have large mean offsets, so the required bias shift is correspondingly large, and bias learning is too slow to make meaningful progress within practical training budgets. Until the bias catches up, dead-by-ReLU features have no mechanism to revive, which is why they plateau. Dictionary size 8192, $k = 64$.

**What drives collateral death.** During threshold collapse, features strongly aligned with $\mu$ initially fire on every input and quickly learn to reconstruct the mean. Their decoder columns are constrained to unit norm, so they compensate by increasing their activation magnitudes. Once these features capture $\mu$, their activations decrease and the TopK threshold drops, and previously excluded features can now enter the top-$k$. But the same downward pressure that opens these slots also pushes some active features' pre-activations below zero, converting them to dead-by-ReLU. Figure S8 makes this visible on individual features. The bottom panels show the TopK threshold (purple) rising as mean-capturing features build up their activations, then collapsing sharply once those features hit ∼100% mean capture (orange) and their activations start to decrease. The top panels (one line per feature) show the consequence: alive features (green) follow the threshold downward, and some borderline features cross below zero and convert to dead-by-ReLU.

Higher $\gamma$ amplifies this effect. Mean-capturing features need to reconstruct the full $\mu$ using unit-norm decoder columns, so they have to fire at activation magnitudes proportional to $\|\mu\|$. At higher $\gamma$, these activations start larger, so when they later decrease (as the bias takes over) the drop is correspondingly larger. The TopK threshold tracks these decreasing activations and falls further, pushing more borderline features below zero.

Bias learning can exacerbate collateral death. As the bias begins to learn $\mu$, it shifts all pre-activations, pushing features that were already borderline (positive but decreasing) further below zero. This effect is visible in Figure 5d and at the per-feature level in Figure S8: the dead-by-ReLU rise during the collapse window is less pronounced when the bias is frozen (panel b) than in the baseline (panel a). The effect is secondary to threshold collapse itself, but it contributes at moderate $\gamma$ where many features hover near zero.

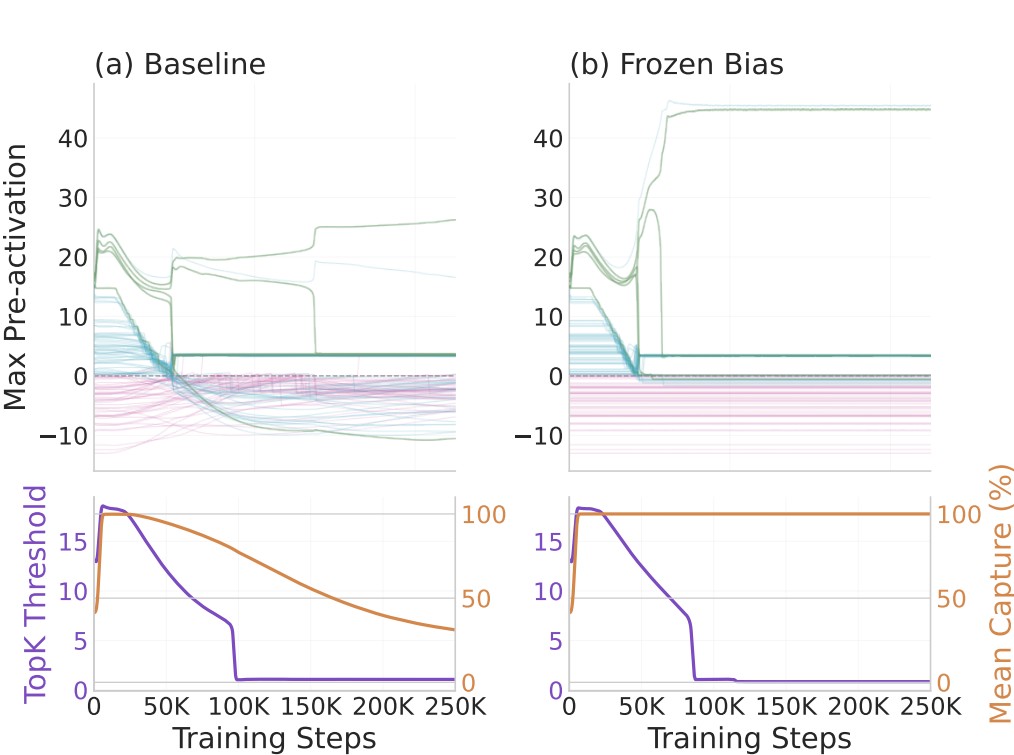

*Figure S8.* **Per-feature pre-activation trajectories at** $\gamma = 20$**.** Each line shows one feature's maximum pre-activation across evaluation inputs over training. Colors indicate initialization status: alive (green), dead-by-TopK (blue), dead-by-ReLU (pink). (a) Baseline training. Dead-by-ReLU features slowly rise toward zero as the bias learns $\mu$. (b) Frozen bias. Dead-by-ReLU features are completely stationary: flat horizontal lines throughout training. Bottom panels show the TopK threshold (purple, left axis) and mean capture percentage (orange, right axis). Threshold collapse coincides with features capturing the mean in both conditions; dead-by-ReLU revival tracks bias learning in the baseline and is absent when the bias is frozen.

**Collateral death and AuxK.** Figure S9 shows the collateral death ratio as a function of $\gamma$, comparing baseline and +AuxK conditions. At moderate $\gamma$ (5–20), AuxK keeps the ratio near zero: it provides gradient to dead-by-TopK features during threshold collapse, helping them stabilize above zero rather than crossing into dead-by-ReLU. At high $\gamma$ ($\geq 25$), AuxK can no longer prevent collateral death: the threshold drops are too steep and the pre-activation shifts too large. Both conditions converge to a ratio near 1.0 at $\gamma = 30$.

This is the mechanism by which AuxK reduces dead-by-ReLU count in Figure 5d, despite providing no gradient to features with negative pre-activations. At moderate $\gamma$, preventing collateral death is the difference between full recovery and permanent death. At high $\gamma$, most dead-by-ReLU features were born dead at initialization, not created as collateral, so this benefit addresses a shrinking fraction of the total problem.

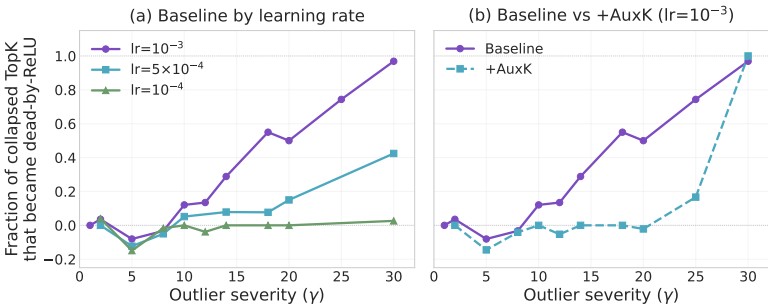

*Figure S9.* **AuxK prevents collateral death at moderate $\gamma$ but not at high $\gamma$.** The collateral death ratio (fraction of collapsed dead-by-TopK features that become dead-by-ReLU during the collapse window) as a function of $\gamma$. Baseline (purple): ratio rises steadily from $\sim 0$ at $\gamma = 2$ to $\sim 1.0$ at $\gamma = 30$. +AuxK (cyan dashed): ratio stays near zero through $\gamma = 20$, then rises sharply to $\sim 1.0$ at $\gamma = 30$. Negative values at low $\gamma$ indicate that some dead-by-ReLU features revived during the collapse window (net decrease rather than increase).

**Learning rate sensitivity.** The exact timing and magnitude of collateral death depend on learning rate. Very low learning rates slow threshold collapse (features take longer to capture the mean, so the threshold drops later and more gradually), reducing collateral death. Very high learning rates produce steeper threshold drops and more collateral death, because large weight updates can overshoot, sending more features below zero in a single step. The qualitative trend (collateral death increasing with $\gamma$) holds across the learning rates we tested.

### C.2. Per-feature view of the two-stage recovery

Figure S8 shows per-feature pre-activation trajectories at $\gamma = 20$ under baseline training (a) and frozen bias (b). Each line is one feature's maximum pre-activation over training, colored by its initial status (alive=green, dead-by-TopK=blue, dead-by-ReLU=pink); bottom panels show the TopK threshold (purple) and mean-capture progress (orange).

**Without bias learning, dead-by-ReLU features stay dead.** The pink features in baseline (a) slowly rise toward zero as the bias learns $\boldsymbol{\mu}$. In frozen bias (b), the same features are flat lines that do not move throughout training. Bias learning is the only mechanism that revives them.

**TopK collapse is bias-independent.** In both panels, the TopK threshold collapses right after mean capture completes (bottom panels), and the blue features jump upward during this window on the same timescale (top panels). Frozen bias does not prevent TopK revival.

**The bias keeps pushing features down after they cross zero.** In baseline (a), features that died during collapse don't stop there. The bias keeps shifting them downward as it learns, so they end up well into negative pre-activation territory where they're slow to revive. In frozen bias (b), descending features stop near zero.

### C.3. Why bias learning is slow

Two mechanisms make bias learning slow. The first is the asymmetric gradient scaling noted in Section 5.2: feature updates pick up a factor of $\|\boldsymbol{\mu}\|$ from the input while bias updates don't. We measure this directly at fixed $\gamma = 20$ with $\|\boldsymbol{\mu}\|$ ranging from 10 to 200; the gradient ratio $|\partial \mathcal{L} / \partial \mathbf{W}_{\text{enc}}[:, 0]| \, / \, |\partial \mathcal{L} / \partial \mathbf{b}_{\text{enc}}|$ at initialization scales linearly with $\|\boldsymbol{\mu}\|$ (Figure S10a). The asymmetry persists under SignSGD (Bernstein et al., 2019), where every parameter receives the same magnitude update: features capture $\boldsymbol{\mu}$ within a few thousand steps at all $\|\boldsymbol{\mu}\|$ values, while bias capture depends strongly on $\|\boldsymbol{\mu}\|$ (at $\|\boldsymbol{\mu}\| = 10$ the bias reaches $\sim 60\%$ after 1M steps; at $\|\boldsymbol{\mu}\| \geq 200$ it barely moves) (Figure S10b). The asymmetry is structural, not an optimizer artifact.

The second mechanism is gradient competition. Initial bias gradients are large because every input has a large residual on the outlier dimension, but features capture $\boldsymbol{\mu}$ within the first $\sim$100K steps. Once they do, the reconstruction residual shrinks and the bias gradient loses its consistent direction. We can isolate the effect by removing the competition: with all features frozen, the bias is the only parameter that can reduce loss on the outlier dimension, and it converges to $\sim$99% of $\boldsymbol{\mu}$ within $\sim$120K steps. Under normal training (features learning in parallel), it reaches only $\sim$87% after 1M steps (Figure S11).

Together, these mechanisms decouple $\gamma$ from $\|\boldsymbol{\mu}\|$: at fixed $\gamma$, the initial death rate is identical, but recovery time scales with

$\|\boldsymbol{\mu}\|$. AlphaFold3 layer 0 and ESM3 layer 9 both have post-LayerNorm $\gamma \approx 15$ and start at $\sim$98% dead, but AlphaFold3's raw $\|\boldsymbol{\mu}\| \approx 80,000$ is $160\times$ larger than ESM3's $\approx 500$. After the same training budget, AlphaFold3 retains 98% dead features while ESM3 drops to 83%.

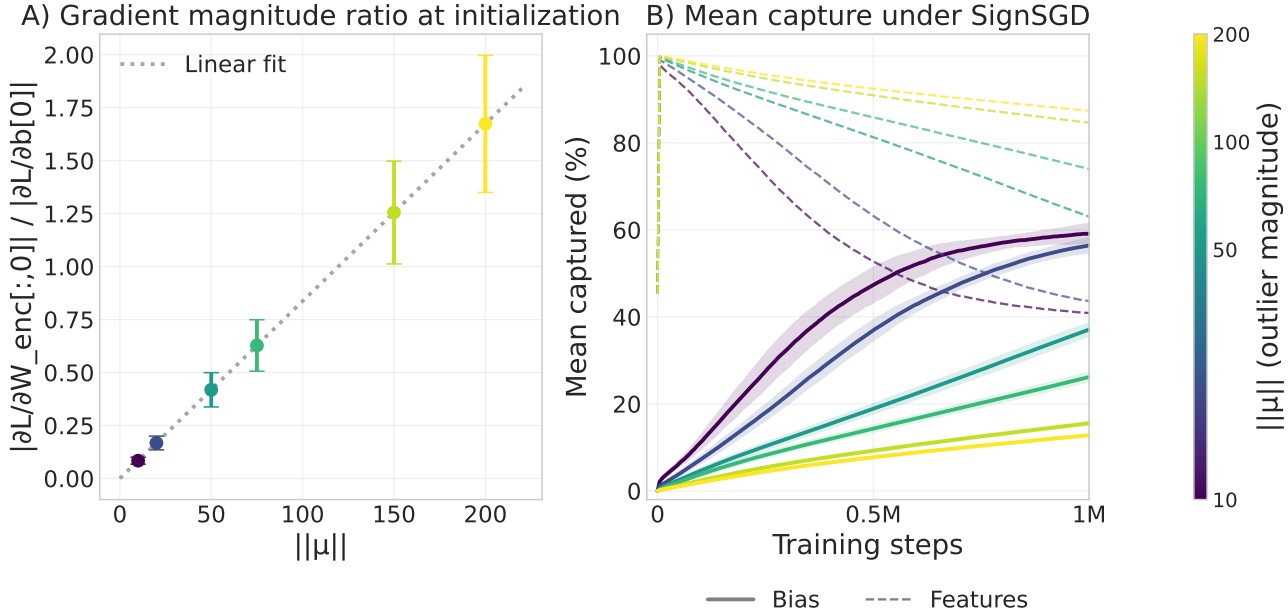

*Figure S10.* **Feature gradients grow with $\|\boldsymbol{\mu}\|$; bias gradients don't.** (a) Gradient magnitude ratio (encoder weights on outlier dimension vs. encoder bias) at initialization, plotted against $\|\boldsymbol{\mu}\|$ at fixed $\gamma = 20$. The ratio scales linearly with $\|\boldsymbol{\mu}\|$. (b) Mean capture under SignSGD (all parameters receive equal magnitude updates). Dashed lines: feature capture (converges within a few thousand steps at all $\|\boldsymbol{\mu}\|$). Solid lines: bias capture (depends strongly on $\|\boldsymbol{\mu}\|$; at $\|\boldsymbol{\mu}\| = 10$ reaches $\sim$60% by 1M steps; at $\|\boldsymbol{\mu}\| \geq 200$ barely moves from zero). All conditions use $\gamma = 20$; colors indicate $\|\boldsymbol{\mu}\|$ (viridis scale, 10 to 200).

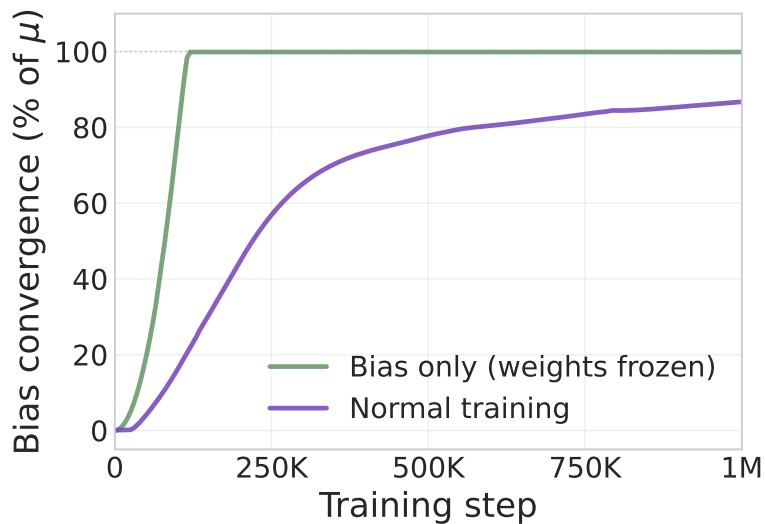

*Figure S11.* **Feature competition slows bias learning.** Bias convergence (fraction of $\boldsymbol{\mu}$ captured by $\mathbf{b}_{\text{dec}}$) at $\gamma = 20$ under two conditions. Green: bias-only training (all weights frozen): the bias converges to $\sim$99% of $\boldsymbol{\mu}$ within $\sim$120K steps. Purple: normal training, where the bias reaches only $\sim$87% after 1M steps, roughly $5\times$ slower. When features capture the mean first, they reduce the reconstruction residual that drives bias learning.

**AuxK does not speed up bias learning.** Across $\gamma \in \{8, 14, 20, 30, 50\}$, the baseline and +AuxK bias convergence trajectories overlap (Figure S12). The gap between $\gamma$ values is far larger than the gap between baseline and +AuxK at any fixed $\gamma$. AuxK's benefit on dead-by-ReLU operates through encoder weight dynamics (stabilizing dead-by-TopK features

above zero during threshold collapse, Appendix C.1), not through accelerating bias learning.

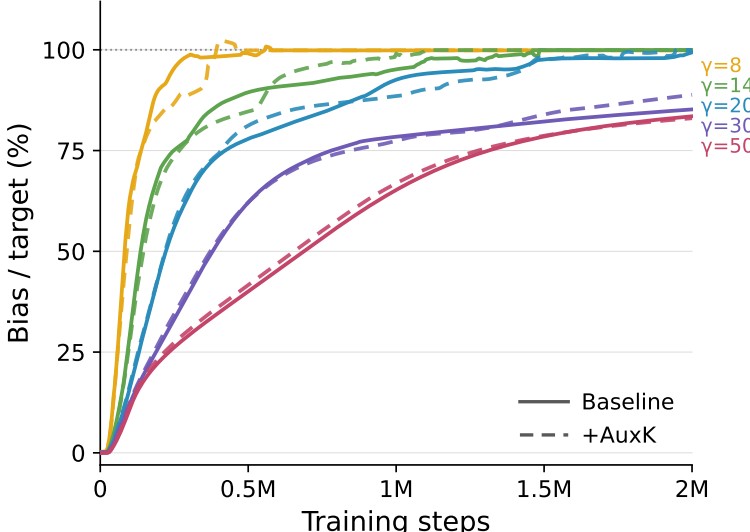

*Figure S12.* **AuxK does not speed up bias learning.** Bias convergence (fraction of $\mu$ captured by $\mathbf{b}_{\text{dec}}$) over training for baseline (solid) and +AuxK (dashed) at $\gamma \in \{8, 14, 20, 30, 50\}$. At each $\gamma$ value, the two conditions follow closely overlapping trajectories. The gap between $\gamma$ values is far larger than the gap between baseline and +AuxK at any given $\gamma$, confirming that outlier severity (not the presence of auxiliary losses) determines the rate of bias learning.

## D. Robustness across learning rates, architectures, and failure modes

### D.1. Mean-centering makes synthetic models more robust across learning rates

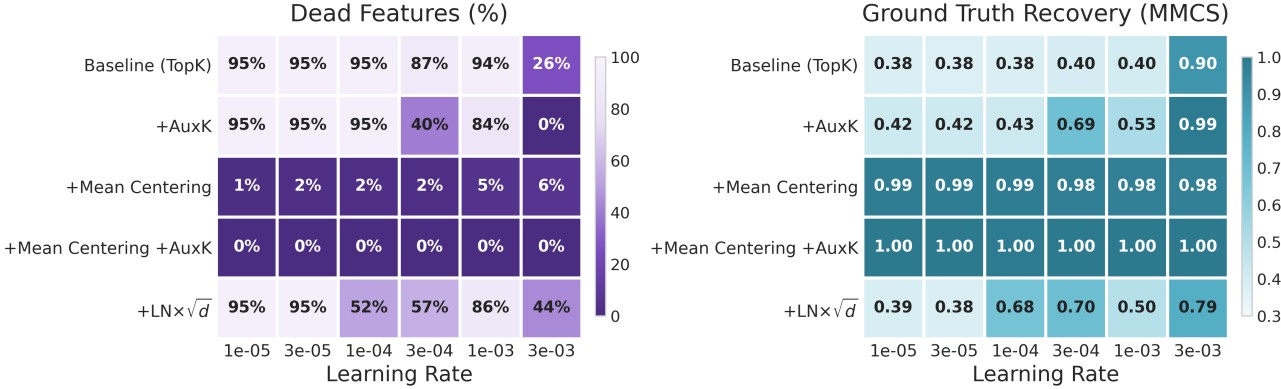

*Figure S13.* **Mean-centering makes synthetic models robust to learning rate across two orders of magnitude.** Dead feature percentage (left) and ground-truth recovery via mean maximum cosine similarity (right) on synthetic data ($\gamma = 40$), varying learning rate and preprocessing. TopK SAEs, dictionary size 8192, $k = 64$. Without mean-centering, the baseline and AuxK both show $> 84\%$ dead features at all but the highest learning rate. Mean-centering brings death below $6\%$ and achieves near-perfect recovery (MMCS $\geq 0.98$) across the full range. Layer normalization with $\sqrt{d}$ rescaling does not resolve death and hurts recovery.

### D.2. Mean-centering eliminates outlier-induced death across models and architectures

**TopK SAEs.** Mean-centering eliminates outlier-induced death consistently across all models in our evaluation suite (Figure S14). Improvements are proportional to outlier severity $\gamma$: high-$\gamma$ models (ESM3, AlphaFold3, ESM2-3B) show the largest reductions, while low-$\gamma$ models (GPT-2, Pythia) show minimal change. One model, Evo1, is an exception whose residual death has a different geometric cause; we analyze it in Appendix E.

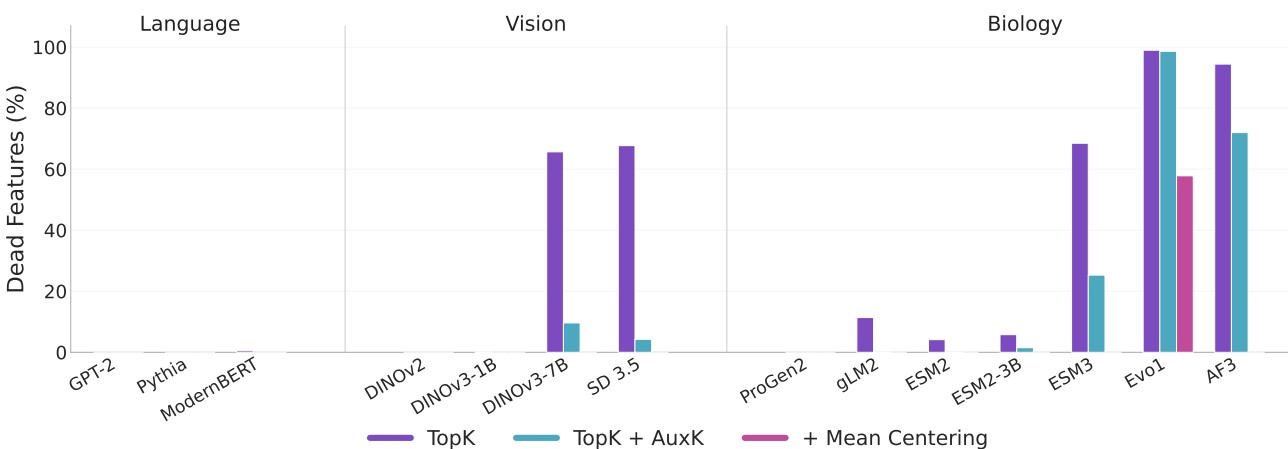

*Figure S14.* **Mean-centering and AuxK each reduce death rates after training across the majority of embedding sources.** Single (middle-of-model) representative layer selected from each model. Dictionary size 8192, $k = 64$, 1M training steps. For each method–model pair we trained SAEs at several learning rates and report the run with the lowest reconstruction MSE.

**ReLU and JumpReLU SAEs.** TopK SAEs enforce exact sparsity by keeping the $k$ largest latent activations and zeroing the rest. ReLU and JumpReLU SAEs work differently: they enforce sparsity indirectly, through their activation functions and penalty terms. A ReLU SAE applies a standard ReLU to the encoder output and adds an $L_1$ penalty on activations to discourage too many features from firing at once. The competition between features is implicit: the $L_1$ cost pressures the model to concentrate mass on fewer features rather than explicitly capping how many can be active. JumpReLU (Rajamanoharan et al., 2024) replaces the smooth ReLU with a hard threshold $\theta$: a latent is exactly zero if its pre-activation falls below $\theta$ and passes through otherwise. Here sparsity is governed by the threshold level (learned or scheduled) rather than a fixed budget, so competition is again indirect: features must clear the threshold to participate at all.

A practical consequence is that the realized $L_0$ (average number of nonzero latents) is harder to control than with TopK. Two runs with different $L_1$ coefficients or threshold schedules can end up at very different $L_0$ values, making apples-to-apples comparison difficult. We target $L_0 \approx 64$ to match our TopK experiments, but for several high-$\gamma$ models our hyperparameter sweep did not land in this range. We provide both the percent features that are dead features, as well as the final sparsity (L0) in Figure S15.

For JumpReLU, mean-centering helps substantially on models with moderate $\gamma$: DINOv2-L drops from 62.8% to 0.0% dead at reliable $L_0$, and SD 3.5 drops from 67.7% to 0%. On more extreme models (Evo1, AF3), JumpReLU+MC achieves 0.0% dead but only by abandoning sparsity entirely ($L_0 > 640$): the soft threshold cannot maintain target sparsity on these inputs even with centering, and TopK's hard selection is needed.

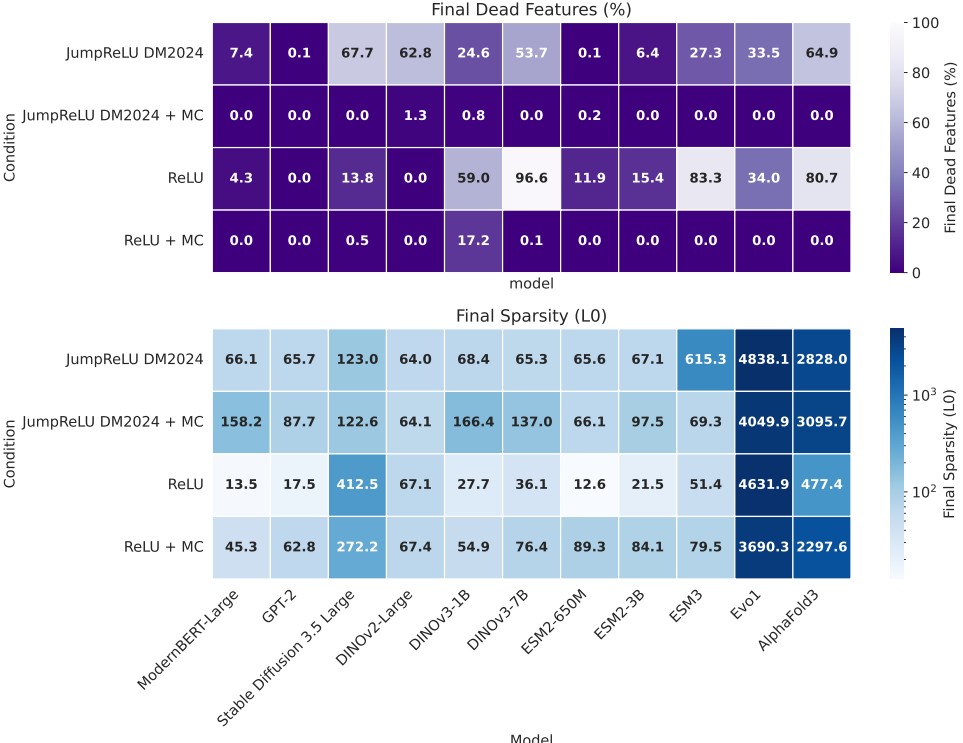

*Figure S15.* **Mean-centering reduces death across SAE architectures, though sparsity control varies.** Top: dead features after 100K steps. Bottom: final $L_0$. On extreme models (Evo1, AlphaFold3), JumpReLU+MC achieves 0% dead only by abandoning target sparsity ($L_0 > 640$). ReLU sweep: $L_1 \in \{10^{-4}\text{–}10^{-1}\}$, best $L_0$ near 64 selected. Dictionary size 8192.

### D.3. Mean-centering increases MonoSemanticity Scores

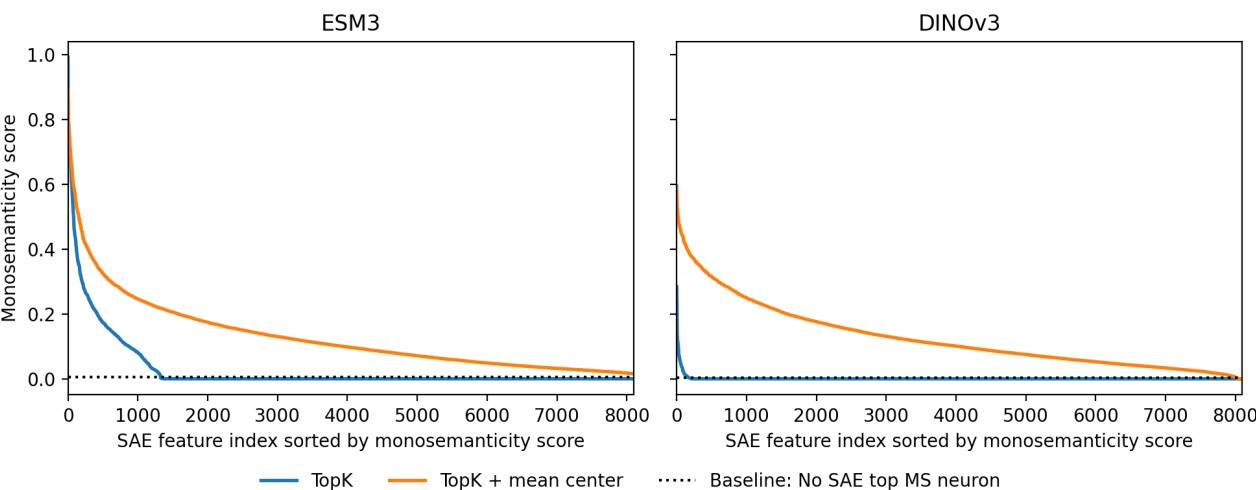

*Figure S16.* **Mean-centered SAEs produce more monosemantic features.** Monosemanticity Scores (Pach et al., 2025) for all 8192 features, sorted by decreasing score. Each feature's MS measures the activation-weighted pairwise similarity of its top-activating inputs under an independent embedding model (ESM2 for ESM3; CLIP-ViT for DINOv3). Blue: baseline TopK. Orange: TopK + mean centering. Dotted: best single raw neuron (no SAE). On ESM3 (left), baseline features drop to zero around index 1500 (matching the ~1400 alive features); mean-centered features maintain nonzero scores across roughly 5000 features. On DINOv3 (right), baseline features have almost no monosemanticity due to high death rates; mean-centering recovers meaningful scores across most of the dictionary. Dictionary size 8192, $k = 64$.

## D.4. Mean-centering does not address training-dynamics sources of death

The geometric analysis in the main text focuses on death that originates at initialization and persists into training. But features can also die from training dynamics alone, even when initialization is death-free. We describe three such mechanisms here.

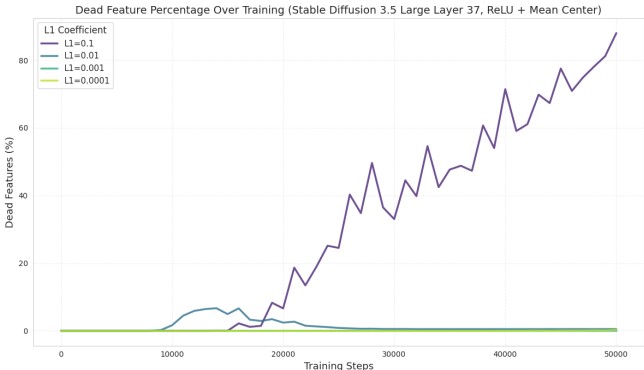

*Figure S17.* **High sparsity pressure causes death during training even from a death-free initialization.** Dead features over training for a mean-centered ReLU SAE (Stable Diffusion 3.5, layer 37) at four $L_1$ values. At $L_1 = 10^{-1}$, death rises to ~80% despite zero geometric death at initialization.

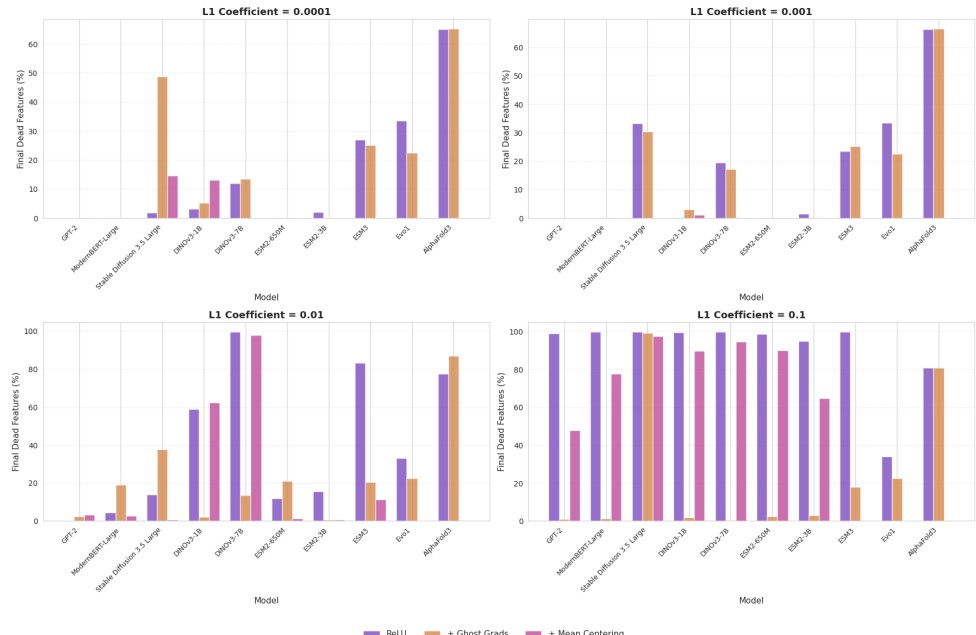

*Figure S18.* **Ghost gradients and mean-centering address different failure modes.** Dead features after 100K steps for ReLU SAEs (baseline, +ghost gradients, +mean-centering) at $L_1 \in \{10^{-4}$–$10^{-1}\}$. Mean-centering fixes geometric death at low $L_1$; ghost gradients partially help with training-dynamics death but not geometric death. Dictionary size 8192.

**High learning rate and high sparsity.** Too-high learning rates can cause features to overshoot early in training and collapse to zero, regardless of initialization. Figure S13 shows this on synthetic data: learning rate alone drives death from near-zero to over 90%, independent of whether geometric preprocessing was applied. High sparsity pressure has a similar effect. In ReLU SAEs, cranking up the $L_1$ coefficient progressively kills features that were initially alive, even when starting from a mean-centered, zero-death initialization (Figure S17). A decreasing-$k$ schedule in TopK SAEs can produce the same outcome.

Ghost gradients (Jermyn & Templeton, 2024), which inject synthetic gradient signal into inactive features, partially help on some models (Figure S18). But they do not fix geometric death from mean offsets, much as AuxK does not on its own. We have not pinned down exactly why, but suspect a related mechanism: ghost gradients scale with the exponential of the

pre-activation value, and when mean offsets push pre-activations far negative, the resulting gradient signal is tiny, likely too small to help the SAE learn to compensate for the offset.

**Stale momentum.** Bricken et al. (2023a) identified a separate optimizer-level failure mode: when a feature stays inactive for many steps, its Adam momentum terms go stale. If the feature then briefly activates, the accumulated stale momentum produces a disproportionately large update that can knock it right back to permanent inactivity. Wang et al. (2025) showed that SparseAdam (a variant that only updates momentum for features with nonzero gradients) substantially reduces this kind of death during SAE training. We do not investigate SparseAdam in the present work, but note it as a complementary approach: mean-centering fixes the geometric conditions that cause death at initialization, while SparseAdam addresses the optimizer dynamics that can kill features over the course of training.

### D.5. Geometric median vs. arithmetic mean for centering

We initialize the SAE pre-encoder bias to the geometric median (GM) of the activation distribution rather than the arithmetic mean (AM). On 15 of 19 models in our suite, GM and AM agree closely and the choice is immaterial. On the remaining four (DINOv3-B, DINOv3-7B, ModernBERT, ModernBERT-Large), a small fraction of outlier tokens with extreme activation norms inflates the AM by $1.7$–$5.5\times$ relative to the GM, and on three of these also rotates it sharply away from the typical token ($\cos(\boldsymbol{\mu}_{\mathrm{AM}}, \boldsymbol{\mu}_{\mathrm{GM}}) = 0.29$–$0.72$). Initializing the bias to this contaminated AM *increases* init-time dead features on three of the four affected models, in some cases worse than no centering at all (Table S7). GM centering eliminates the problem ($\leq 2.3\%$ dead across all four).

The GM minimizes $\sum_i \|\mathbf{x}_i - \boldsymbol{\mu}\|_2$ rather than the sum of squared distances, giving it a breakdown point of $50\%$: up to half of token activations can be arbitrarily corrupted without affecting the estimate. Heavy-tailed token norms are common in vision transformers and recent encoder language models, where attention sinks and register tokens produce a small set of high-norm outliers. Trimmed means and the coordinate-wise median yield essentially identical results to GM on these models; we choose GM because it is affine-equivariant and requires no choice of trimming fraction. We compute it via Weiszfeld's algorithm on a calibration set of 500 tokens.

| Model | Layer | Dead % (no center) | Dead % (AM) | Dead % (GM) |
|---|---|---|---|---|
| DINOv3-B | 3 | 57.6 | 91.7 | **0.0** |
| DINOv3-7B | 4 | 87.6 | 59.0 | **2.3** |
| ModernBERT | 16 | 55.1 | 89.0 | **0.0** |
| ModernBERT-Large | 20 | 34.5 | 88.3 | **0.0** |

*Table S7.* **Geometric median centering eliminates outlier-induced death on models where arithmetic mean centering makes it worse.** TopK SAE, dictionary size 8192, $k = 64$, evaluated at random initialization on 100K held-out tokens. On three of four affected models, AM centering produces more dead features than no centering at all; GM centering brings dead-by-TopK below 2.5% across all four. Layers shown are the layer with the largest gap between AM-centered and GM-centered death rates for each model.

### E. Features die when activation variance concentrates in too few directions

Mean-centering eliminates outlier-induced death across models (Section 6, Figure S14), but on some models and layers substantial death remains at initialization even after centering. This remaining death is entirely dead-by-TopK: centering removed the mean offset, so no feature has permanently negative pre-activations, but many features still never rank in the top $k$ on any input.

At the representative mid-depth layers we train our models on (Figure S14), Evo1 is our only outlier where substantial death survives mean-centering. Centering reduces init-time death from 96% to 73% but does not eliminate it, and the residual does not recover during training ($\sim$58% dead-by-TopK at the end). The fact that this remaining death is, like the outlier-induced death, present before any training implies that it is another quirk of the geometry of our activations.

When we look at post-centering death rates across every layer of every model at initialization, several other models (DINOv3-7B, ESM2-3B, ESM3, and ProGen2-base) also have nonzero post-centering death in a few transformer layers. These death rates are much smaller than Evo1's (20–25%) so easier to recover to near zero during training and also primarily present in the earliest layers of the (typically worst at layer 1) (Table S8), however they highlight that this phenomenon is not specific to Evo1 and provide more examples by which to study it.

We find that the common factor in each of these layers, most extreme in Evo1, is that activation variance is highly concentrated in a small number of principal components (PCs, the eigenvectors of the activation covariance; their variances are the corresponding eigenvalues). When a few PCs carry nearly all the variance, features that project onto them dominate the TopK competition on every input, and the rest never fire. We quantify this concentration with the *effective rank* of the covariance, $\exp(H(p_1, \ldots, p_d))$ where $p_i = \lambda_i / \sum_j \lambda_j$ is the normalized eigenvalue distribution ([Roy & Vetterli, 2007](#)); this is low when variance is concentrated in few PCs and high when it is spread across many. As with our other metrics, we calculate this on post-LayerNorm activations to factor out per-token magnitude differences, ensuring the concentration we measure is directional rather than scale-driven.

### E.1. Variance concentration causes a small group of features to monopolize the TopK competition

This concentration of variance in a few PCs creates unequal competition among features, producing dead-by-TopK features. TopK selects the $k$ features with the largest pre-activations on each input, so a feature only fires if its pre-activation sometimes exceeds those of at least $n - k$ others. Each feature's pre-activation is a dot product $\mathbf{w}_i^\top \mathbf{x}$, so its variance across inputs is $\mathbf{w}_i^\top \Sigma \mathbf{w}_i$ for activation covariance $\Sigma$. This is the activation variance along each PC of $\Sigma$, weighted by $\mathbf{w}_i$'s alignment with each PC.

When the per-PC variances are roughly equal, every random weight vector captures similar total variance and features compete fairly. When the top few PCs carry far more variance than the rest, only weight vectors aligned with those top PCs capture meaningful variance; the rest produce low-variance pre-activations and rarely rank in the top $k$. The few features that happen to align monopolize the TopK competition on every input, and the rest are dead-by-TopK.

We visualize this phenomenon using the activations of Evo1, where the effect is most severe.

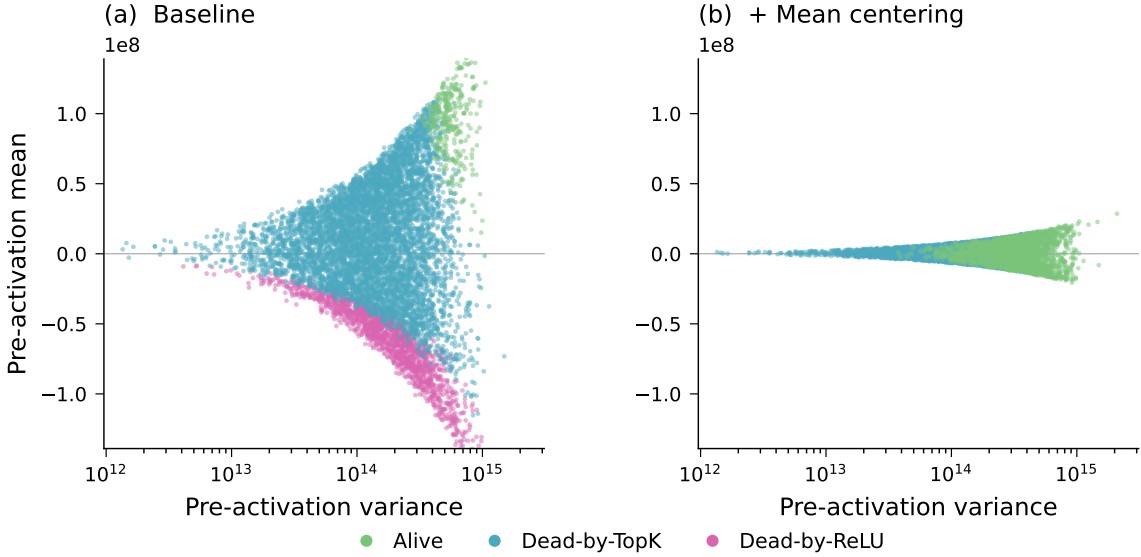

*Figure S19.* **After mean-centering on Evo1, a minority of high-variance features monopolize the TopK competition. (a)** Baseline (no preprocessing): pre-activation mean vs. pre-activation variance, colored by death type. 17% dead-by-ReLU and 79% dead-by-TopK. **(b)** Mean-centered: dead-by-ReLU disappears entirely. 73% dead-by-TopK remains. Dictionary size 8192, $k = 64$, evaluated at initialization on 100K held-out samples.

In Figure S19, we see that mean-centering removes the mean offset that produced dead-by-ReLU features, yet 71% of features remain dead-by-TopK, clustered in the low-variance region of the scatter. The per-feature pre-activation variances split sharply: alive features have $2.9\times$ higher median variance than dead features.

However, variance alone does not fully predict which features die: the split is not a sharp cutoff. When most features' pre-activations are driven by the same few underlying signals, a lower-variance feature that tracks the same pattern as a dominant one, just with smaller amplitude, peaks when the dominant one peaks and never outranks it. Some moderate-variance features die for this reason; some low-variance features survive by peaking on inputs where the dominant features are middling.

### E.2. Removing variance concentration removes the death

If variance concentration causes the unequal competition, equalizing variance should eliminate death. We evaluate two interventions: PCA whitening (rotates and rescales the data) and Active Subspace Initialization (changes only the SAE weights, leaving the data untouched).

**Equalizing the per-PC variances with PCA whitening eliminates the death.** PCA whitening rotates activations into the PC basis and rescales each component to unit variance. The resulting covariance is the identity: every direction carries the same variance, so random weight vectors capture the same expected pre-activation variance regardless of orientation.

To understand how strongly eigenvalue concentration drives the death, we smoothly interpolate between the original and equalized spectra. Rescaling each eigenvalue $\lambda_j$ to $\lambda_j^{1-\alpha}$ for $\alpha \in [0, 1]$ continuously deforms the spectrum from $\alpha = 0$ (mean-centering only) to $\alpha = 1$ (full PCA whitening). On Evo1 layer 14, dead-by-TopK features decrease monotonically with $\alpha$ and reach zero by $\alpha \approx 0.6$ (Figure S20).

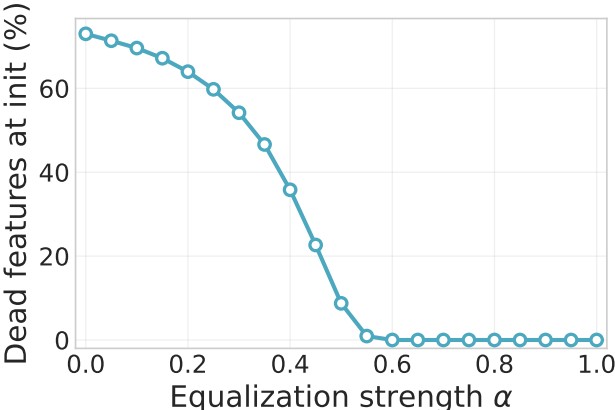

*Figure S20.* **Smoothly equalizing the eigenvalue spectrum of Evo1 layer 14 activations monotonically reduces dead features.** Each eigenvalue $\lambda_j$ is rescaled to $\lambda_j^{1-\alpha}$: $\alpha = 0$ leaves the spectrum unchanged (mean-centering only), $\alpha = 1$ sets all eigenvalues to 1 (full PCA whitening). Death drops from 73% to 0% by $\alpha \approx 0.6$. TopK SAE, dictionary size 8192, $k = 64$, evaluated at random initialization on 100K held-out tokens.

At $\alpha = 1$, PCA whitening reaches 0% dead features at initialization on every model in our suite (Table S8). Z-scoring (rescaling each coordinate to unit variance) does not fix the problem: it touches only the diagonal of the covariance, leaving the off-diagonal correlations that produce the eigenvalue concentration intact. On Evo1, Z-scoring reduces death from 73% to 49% but does not reach zero.

Whitening is the simplest universal fix, but it changes the loss function: dimensions that originally carried $1000\times$ more variance now contribute equally to MSE, forcing the SAE to represent formerly low-variance directions as faithfully as high-variance ones. Saraswatula & Klindt (2025) found that this reweighting improves feature quality on SAEBench. But high-variance directions may carry more of the structure worth decomposing, so a less invasive fix is worth pursuing.

**Active Subspace Initialization fixes the remaining cases without changing the data.** Wang et al. (2025) observed a similar phenomenon of variance concentration in the outputs of transformer attention blocks: most of the variance lies along a small number of PCs resulting in low rank output matrices that have dead features when SAEs are trained on them, the same pattern we see in our extreme cases. They propose Active Subspace Initialization (ASI): project encoder and decoder weights onto the top $d_{\text{init}}$ principal components of the activations, where $d_{\text{init}}$ is a hyperparameter that controls how many PCs are used (they find their method is robust to its choice). This places features into the high-variance subspace at initialization. Because the data is unchanged, the loss function is identical to standard training.

The idea is that if features start in comparable-variance directions, the TopK competition begins fair even though the underlying spectrum is concentrated. This works in Wang et al.'s setting, where the attention outputs in the models they study have effective ranks of 1000–2400 and many top PCs carry comparable variance.

However, our activations are far lower-rank: in the extreme cases, one or two top PCs carry nearly all the variance. Any $d_{\text{init}}$ in Wang et al.'s recommended range (their default is 768, far larger than our effective ranks of 3–31 on the affected layers)

| Model | Layer | Eff. rank / $d$ (%) | Raw dead (%) | MC dead (%) | PCA dead (%) | Fix |
|---|---|---|---|---|---|---|
| Evo1 | 14 | 0.07 | 95.9 | **73.0** | 0.0 | **PCA** |
| DINOv3-7B | 1 | 0.12 | 74.6 | **23.5** | 0.0 | **PCA** |
| ESM2-3B | 1 | 1.22 | 98.1 | **25.7** | 0.0 | **PCA** |
| ESM3 | 1 | 1.44 | 98.9 | **20.8** | 0.0 | **PCA** |
| ProGen2-base | 1 | 1.44 | 59.5 | **22.6** | 0.0 | **PCA** |
| DINOv2-L | 1 | 1.30 | 55.8 | 2.3 | 0.0 | MC |
| SD3.5-L | 4 | 1.80 | 61.4 | 0.0 | 0.0 | MC |
| ProGen2-Large | 1 | 1.85 | 51.0 | 0.5 | 0.0 | MC |
| DINOv3-B | 1 | 2.85 | 86.4 | 1.0 | 0.0 | MC |
| DINOv2-B | 1 | 5.43 | 54.9 | 0.0 | 0.0 | MC |
| ESM2-650M | 1 | 7.92 | 91.6 | 0.0 | 0.0 | MC |
| AlphaFold3 | Pairformer | 8.13 | 99.0 | 1.0 | 0.0 | MC |
| ESM2-35M | 1 | 10.41 | 82.0 | 0.0 | 0.0 | MC |
| GLM2 | 27 | 19.70 | 49.0 | 0.0 | 0.0 | MC |
| ModernBERT-L | 25 | 19.75 | 51.3 | 0.0 | 0.0 | MC |
| Pythia-410M | 18 | 31.09 | 10.3 | 0.0 | 0.0 | MC |
| Pythia-70M | 2 | 37.49 | 3.6 | 0.0 | 0.0 | MC |
| ModernBERT | 5 | 37.63 | 67.5 | 0.0 | 0.0 | MC |
| GPT-2 | 11 | 45.53 | 32.7 | 0.0 | 0.0 | MC |

*Table S8.* **Per-model dead-feature rates at the worst (highest MC dead) layer.** Raw, MC, and PCA columns are dead-by-TopK percentages at initialization. Effective rank is computed post-LayerNorm and normalized by hidden dimension $d$. The **Fix** column gives the recommended preprocessing per model: MC when post-MC death is below 5%, PCA otherwise.

still leaves a single dominant PC in the projected subspace, and ASI does not meaningfully reduce dead features.

The only $d_{init}$ values that produce low dead rates on Evo1 are 1 or 2, where ASI is degenerate: at $d_{init} = 1$ every SAE feature is essentially $\pm PC_1$, and at $d_{init} = 2$ all features live in a 2D plane regardless of dictionary size. A high-capacity dictionary collapses to a 1D or 2D subspace at the start of training, negating the benefit of using a large dictionary.

**Practical guidance.** Most models reach near-zero dead features under mean-centering alone (Table S8). When the post-LN effective rank of the activation covariance is below $\sim 2\%$ of the hidden dimension, centering leaves substantial residual death and the activations are intrinsically low-rank; PCA whitening reaches 0% in every such case. ASI is an alternative when changing the loss is undesirable, with the caveat that small $d_{init}$ (on the order of the effective rank of the affected layers) is needed since larger values reintroduce the variance hierarchy. All diagnostics and fixes require only a single pass over a batch of activations.

