# OpenReview forum: "On the Relationship Between Activation Outliers and Feature Death in Sparse Autoencoders"
_ICML.cc/2026/Conference — ICML 2026 regular_

### Official Review · Reviewer_t78L · 2026-02-15

**Soundness:** 2
**Presentation:** 3
**Significance:** 2
**Originality:** 2
**Overall Recommendation:** 4
**Confidence:** 4

**Summary:**

The paper studies why sparse autoencoders often learn many “dead” features that never activate, despite identical training setups across models. It shows that dimension-level activation outliers create input-independent shifts in feature pre-activations at initialization. Then the author propose $\gamma$ to predict the death rate

**Compliance With Llm Reviewing Policy:**

Affirmed.

**Final Justification:**

The authors addressed nearly all of my concerns, and I will raise my score to 4.

**Key Questions For Authors:**

- why genomic model have high outlier in begining, can you provide potential explaination? is your observation similar as observed in GERM (ICML25)
- I am not totally agree with 50%, it is hard to say the disturbution is normal. So the 50% might have some issue
- Whether $\mathrm{Softmax}_1$ can help solve this issue?

**Limitations:**

yes

**Strengths And Weaknesses:**

**Soundness:**
- *A:*
  - The paper distinguishes two concrete death pathways (dead-by-ReLU vs dead-by-TopK) and ties them to different recovery mechanisms
- *D:*
  - Though dead-by-TopK is explain use experiment, but still a little confuse.
  - the experiment still looks like a proof-of-concept experiment, I hope the experiment can be more comprehensive  I provide some suggestion:
    - I suggest you can extend the model to more kind, such as Text (BERT), time-series model (AutoFormer), Genomic (DNABERT-2), Speech (Voxtlm)
    - On the same activation set, do controlled interventions: inject mean shift only (fix variance) and change variance only (fix mean) Repeat across different sparsities (vary $k$) and plot a phase diagram showing how dead-by-ReLU vs dead-by-TopK depend on $\gamma$ and $k$.
    - Go beyond probes/F1 and include metrics closer to SAE’s real purpose: such as avg kursosis, max infitate norm

**Presentation:**
- *A:*
  - the paper structure is clear and use observation to prove the paper discussioin
- *D:*
  - The paper miss compare with some popular activation outlier paper, such as OutEffHop (ICML24), Quantizable Transformers (NIPS23)
  - mention the $\gamma$ in sec 4 again can help people understand

**Significance:**
- *A:*
  - the paper explain the death feature is realted the activation outlier and data used in SAE
- *D:*
    - The paper show not good at EVO1, whether can provide a method to overcome it based on exsiting outlier removal technology

**Originality:**
- *A:*
  - the author use observation to observe the differenet mobility issue of dead features.
  - explain feature death as primarily a data geometry rather than training dyanamic issue
- *D:*
  - the $\gamma$ defination is pretty similar with avg kursois
For a variable \(x\), the kurtosis is
$$
\text{kurt}(x)=\frac{\mathbb{E}\big[(x-\mu)^4\big]}{\sigma^4},
$$

I think the main function of those two fomula is similar

---

> ### Author Rebuttal · Authors · 2026-03-31
>
> Thank you — your suggestions around the TopK theory, controlled interventions, and feature quality have led to substantial new experiments that strengthen the paper. All new figures are available at [anonymous.4open.science/r/sae](https://anonymous.4open.science/r/sae).
>
> ## Unified theory for dead-by-TopK
> We've added a derivation unifying both pathways. Dead-by-ReLU and dead-by-TopK are the same mechanism with different thresholds: ReLU asks whether a feature's shift pushes it below zero, TopK asks whether it beats the k-th ranked feature. At high γ the shifts dominate, giving P(dead, TopK) ≈ Φ(t_k − C/γ), with the ReLU formula as the special case t_k = 0. `topk.pdf` shows this theory curve overlaid on synthetic and real data.
>
> ## Controlled interventions across sparsities
> We added plots sweeping sparsity and width, and isolating mean from variance:
>
> - `width.pdf` shows the phase diagram across k (5–100) and SAE width (10×–100×): γ predicts death rates for both death pathways across all combinations.
> - `variance.pdf` isolates the two components: fixing μ and varying σ, then the reverse. Death tracks γ, not μ or σ individually.
>
> The relationship holds across all tested sparsity levels and when mean is fixed but variance is controlled, confirming γ is the operative quantity.
>
> ## γ vs. kurtosis
> γ measures the mean offset that causes death. Kurtosis is computed on deviations from the mean, so it is blind to the offset. Empirically: γ gives Spearman ρ > 0.9 for both death pathways across 275 model-layer combinations; kurtosis gives ρ < 0.15. γ also points directly to a fix (mean-centering removes μ, sending γ toward zero) while kurtosis offers no analogous intervention. We will reintroduce γ explicitly in Section 4 as suggested.
>
> ## Feature quality metrics
> Following your suggestion to go beyond probes, we computed kurtosis of per-feature activation distributions on four high-death models (`kurtosis.pdf`). Mean-centering (MC) increases median kurtosis by 1.5–15× across all four (SD3.5, DINOv3-7B, ESM3, AF3), indicating more peaked, selective activations. This is computed over alive features only, so the baseline retains only its most selective features — making the MC improvement more meaningful.
>
> Further, we added Monosemanticity Scores on DINOv3 and ESM3 (Pach et al., 2025), measuring whether individual features respond to semantically coherent inputs via pairwise embedding similarity with independent models. `MS_score.pdf` shows the distributions. MC SAEs have far more features with meaningful monosemanticity, and even restricting to alive features only, MC features score higher at every quantile. With MMCS on synthetic data and Swiss-Prot alignment on ESM3, three evaluations confirm MC improves both dictionary capacity and feature quality.
>
> ## 50% limit
> The 50% limit does not require Gaussianity. It follows from initialization symmetry: encoder weights are uniform on the unit sphere, so **w**·**μ** is symmetric around zero — exactly half align positively, half negatively. As γ → ∞, input signal vanishes relative to the shift and the rate converges to 50%.
>
> ## Related work and outlier origins
> These papers study a complementary question: *why* models develop activation outliers vs. *what* those outliers do to SAE training. OutEffHop, GERM, and Quantizable Transformers characterize outlier origins and propose architectural fixes (e.g., Softmax₁ to suppress attention-mechanism outliers). We characterize the downstream consequence for SAEs and provide post-hoc diagnostics for pretrained models. GERM documents severe dimension-level outliers in genomic models — the same pathology that produces high γ in our framework. Their account answers your question about why genomic models have high outliers from the start. Softmax₁ targets attention-mechanism outliers; our outliers are dimension-level activation offsets present even in non-attention architectures (e.g., Evo1's Hyena layers), requiring a different intervention. We will add these to related work.
>
> ## Evo1
> You asked whether existing outlier removal methods could address Evo1 — they can. PCA whitening reduces dead features from 66.6% to 0%. Evo1's activations are extremely low-dimensional (4 of 4096 PCs carry 99% of variance). After centering, no features are stuck below zero, but features aligned with dominant PCs always win TopK. Partial eigenvalue equalization reaches 0% at α = 0.75 (`pca.pdf`). A single-forward-pass eigenvalue check identifies which preprocessing each model needs (`preprocess.pdf`).
>
> ## Model breadth
> We have added gLM2 to strengthen genomics coverage. The diagnostic now spans many model-layer combinations from 10+ model families across five domains (language, vision, protein sequence / structure, genomics), including both transformer and non-transformer architectures. γ predicts death rates across all of them, and since it depends on activation statistics rather than architecture, we expect this to hold in other domains as well.

---

> > ### Author Rebuttal · Reviewer_t78L · 2026-03-31
> >
> > Thank you for the substantial new experiments; they strengthen the paper, especially the controlled sparsity sweeps and the broader model coverage. That said, I am still not fully convinced by the γ vs. kurtosis argument: kurtosis is precisely a statistic for heavy-tailed or outlier-prone activations, so I would like a sharper explanation of why it fails here and why the mean-offset term / λ-like shift is the true causal factor instead; also, please include the full anonymous links for each figure or PDF directly in the rebuttal, because I could not access the referenced files from the current text.

---

> > > ### Author Response · Authors · 2026-04-07
> > >
> > > Thank you for your follow up and for acknowledging that our paper is strengthened!
> > >
> > > ## γ vs. kurtosis
> > > The core issue is that SAE feature death is driven by a slightly different type of outlier dimensions than the typical outliers studied/quantified in the quantization-related literature, and kurtosis is blind to the specific property we care about. We constructed synthetic activations to isolate this cleanly: (https://anonymous.4open.science/r/sae/constant_vs_occasional.pdf). Both panels have the same outlier dimension at the same magnitude; the only difference is whether the outlier is constant (large on every token; the type that we focus on) or occasional (~5% of tokens spike; the type more commonly studied). The constant outlier gives γ ≈ 6, κ = 0.0, and 32% dead features. The occasional outlier gives γ = 0.0, κ ≈ 3, and 0% dead features. Only γ tracks death.
> > >
> > > Two real examples from our data show this same pattern (https://anonymous.4open.science/r/sae/constant_vs_occasional_real.pdf): ESM3 Layer 1 has κ ≈ 0 (perfectly Gaussian deviations) yet 46% dead features, because one dimension sits at has a consistently large negative value on every token (causing large γ). Conversely, DINOv3-1B Layer 7 has κ ≈ 68 (extremely heavy-tailed) but 0% dead features, because its outliers are occasional spikes rather than constant offsets.
> > >
> > > This holds across all 275 real model-layer combinations. As shown in the paper, γ tracks has a strong correlation with both empirical death rates and the curve of predicted death rates from theory; in (https://anonymous.4open.science/r/sae/gamma_vs_kurtosis.pdf) we see that kurtosis does not correlate strongly with empirical death rates (Spearman ρ = 0.10 against dead-by-ReLU).
> > >
> > > The reason kurtosis misses this is that it's computed on deviations from the mean (κ = E[(x−μ)⁴]/E[(x−μ)²]²), so it captures occasional spikes but ignores constant offsets. γ measures how large the mean is relative to per-token variation — exactly the quantity in the shift term **w**_i · **μ** that predetermines feature fate.
> > >
> > > To be clear about the contribution: γ isn't meant to _compete_ with kurtosis as a general outlier metric. The finding is that constant dimension-level offsets — a form of outlier structure that the quantization literature hasn't needed to distinguish from occasional spikes — are what cause feature death. We defined γ because no existing measure captured this; the mechanistic link to feature fate is the contribution, and γ is the tool for quantifying it.
> > >
> > > None of this means kurtosis is wrong for most outlier-prone activations in the quantization setting — GERM itself uses kurtosis reduction as a key metric, and successfully so, because occasional large activations are what destroy quantization precision. SAE feature death is a different downstream consequence driven by the constant offset rather than the tail behavior. GERM and OutEffHop document and address the _causes_ of outlier features, which is upstream of both types of outliers; our work identifies which property of those outliers matters for SAEs specifically.
> > >
> > > We agree this is an important distinction to make and will add this framing along with the comparison figures to the revision.
> > >
> > > ----
> > >
> > > Across both rounds, we have now addressed all points raised in your review: the unified TopK derivation, controlled interventions across sparsities, feature quality metrics beyond probes, the Evo1 preprocessing fix, broader model coverage, and now the direct γ-vs-kurtosis comparison. Given this, and how including the suggested experiments has improved the paper, we would greatly appreciate your reconsidering the score.
> > >
> > > ## Figure links
> > >
> > > All figures are at https://anonymous.4open.science/r/sae/ —direct links:
> > >
> > > - https://anonymous.4open.science/r/sae/topk.pdf
> > > - https://anonymous.4open.science/r/sae/width.pdf
> > > - https://anonymous.4open.science/r/sae/variance.pdf
> > > - https://anonymous.4open.science/r/sae/kurtosis.pdf
> > > - https://anonymous.4open.science/r/sae/MS_score.pdf
> > > - https://anonymous.4open.science/r/sae/pca.pdf
> > > - https://anonymous.4open.science/r/sae/preprocess.pdf
> > > - https://anonymous.4open.science/r/sae/constant_vs_occasional.pdf
> > > - https://anonymous.4open.science/r/sae/constant_vs_occasional_real.pdf
> > > - https://anonymous.4open.science/r/sae/gamma_vs_kurtosis.pdf

---

### Official Review · Reviewer_n7SC · 2026-03-12

**Soundness:** 4
**Presentation:** 3
**Significance:** 3
**Originality:** 3
**Overall Recommendation:** 5
**Confidence:** 3

**Summary:**

The paper looks at feature death in SAEs used for interpretability. It shows that activation outliers in model features can create large mean offsets. As a result, some SAE features always activate while others never do. The paper introduces a simple diagnostic metric to measure this effect and shows that mean-centering at initialization can significantly reduce feature death.

**Compliance With Llm Reviewing Policy:**

Affirmed.

**Final Justification:**

I would prefer to keep my rating unchanged.

**Key Questions For Authors:**

See weaknesses.

**Limitations:**

yes

**Strengths And Weaknesses:**

Pros:
- The paper is well organized and easy to follow.
- It studies a clear and important issue in SAEs: dead features.
- The explanation based on dimension-level activation outliers is intuitive and well motivated.
- The diagnostic metric is simple and easy to compute.
- The empirical study covers many models and modalities, including language, vision, and protein models.
- The proposed fix is simple and practical.

Cons:
- The solution is quite simple and may feel more like an engineering tweak than a new learning method.
- The analysis mainly focuses on initialization; it is less clear how training dynamics interact with the proposed mechanism.
- The practical impact on downstream interpretability or applications is not explored in depth.

---

> ### Author Rebuttal · Authors · 2026-03-31
>
> Thank you for the detailed and supportive review. We're glad the mechanism and diagnostic feel well-motivated, and we appreciate the candid framing around engineering tweak vs. new method — it helped us sharpen our articulation of the contribution. In response to your suggestions, we have run new interpretability experiments whose results are available as new figures at [https://anonymous.4open.science/r/sae](https://anonymous.4open.science/r/sae). We address each point below.
>
> ## Practical impact on interpretability
>
> We have added Monosemanticity Score evaluations on both DINOv3 and ESM3 ([Pach et al., 2025](https://openreview.net/forum?id=DaNnkQJSQf)), measuring whether individual features respond to semantically coherent inputs via pairwise embedding similarity with independent models (CLIP-ViT and ESM2 respectively). `MS_score.pdf` shows the full distributions.
>
> The practical upshot: mean-centered SAEs produce features that are individually more interpretable, not just more numerous. Comparing only alive features, removing the trivial effect of having a larger active dictionary, MC features are _still_ more monosemantic at every quantile, in both models. This is a meaningful finding because one might expect that having fewer alive features would concentrate representational capacity and produce *better* survivors; instead, geometric death forces survivors into superposition and degrades their quality. Raw neuron activations (no SAE baseline) are flat near zero, confirming SAEs perform real decomposition and MC SAEs do it more effectively.
>
> Combined with MMCS on synthetic data and Swiss-Prot concept alignment on ESM3, three independent evaluations across three domains show mean-centering improves both dictionary capacity and the quality of individual features.
>
> ## Engineering tweak vs. new method
>
> We agree with this framing: preprocessing methods like mean-centering, LayerNorm, and PCA whitening are not new. The contribution is understanding *when* each is necessary and *why* they work. Before this paper, centering was a preprocessing choice some teams used and others didn't, with no principled basis for the decision. This matters more now that SAEs are being applied beyond language models to vision, protein, and genomic models, where activation geometry varies widely and practitioners can't rely on defaults tuned for GPT-2. Our analysis provides that basis: γ tells you whether mean offsets are the problem, and for models where centering alone doesn't suffice (e.g., Evo1), examining the eigenvalue spectrum can reveal whether low-rank structure or per-token norm variation requires stronger preprocessing. Both checks require only a single forward pass, letting practitioners triage which preprocessing a new model needs before committing to a training run. `preprocess.pdf` shows the recommended fix for each of 19 models.
>
> ## Analysis focuses on initialization
>
> This is deliberate. At initialization the geometry is cleanest: feature fate is determined by alignment with the activation mean, full stop. During training, death has additional causes (gradient starvation, optimizer dynamics, threshold interactions in TopK) that are harder to disentangle. We do characterize training dynamics in Section 5, where we show that recovery proceeds in two stages even without auxiliary losses. We have also added `dynamics.pdf`, which shows the dead-by-TopK / dead-by-ReLU breakdown over training for three real models (AlphaFold3, Stable Diffusion 3.5, ESM3), confirming the two-phase pattern on real data. But the practical argument for focusing on initialization is simple: if you prevent geometric death at init, the model doesn't have to spend training steps undoing it, and convergence is faster as a result (`lr.pdf` shows this across learning rates from 1e-5 to 3e-3).

---

> > ### Author Rebuttal · Reviewer_n7SC · 2026-04-03
> >
> > I appreciate the authors' efforts in addressing my concerns. I would prefer to keep my rating unchanged.

---

> > > ### Author Response · Authors · 2026-04-07
> > >
> > > Thank you for the thoughtful review and for confirming that your concerns have been addressed!

---

### Official Review · Reviewer_VFKD · 2026-03-13

**Soundness:** 3
**Presentation:** 3
**Significance:** 3
**Originality:** 3
**Overall Recommendation:** 3
**Confidence:** 3

**Summary:**

This paper studies why sparse autoencoders (SAEs) often have dead features. The authors argue that dimension-level activation outliers create input-independent shifts at initialization, so many features become dead before learning starts. They propose a simple diagnostic, show it strongly correlates with dead-feature rates across many model-layer pairs, explain two death modes (dead-by-ReLU and dead-by-TopK), and show that mean-centering largely removes this problem.

**Compliance With Llm Reviewing Policy:**

Affirmed.

**Final Justification:**

I will keep my original score unchanged

**Key Questions For Authors:**

1. The paper provides a clean theoretical account for dead-by-ReLU, but dead-by-TopK is analyzed mainly empirically. Could the authors strengthen the theoretical understanding of the TopK pathway, or clarify more precisely which claims are theoretical versus empirical?
2. How robust are the γ diagnostic and the centering intervention to changes in SAE width, sparsity level k, optimizer settings, and training budget?
3. When should practitioners use mean-centering versus stronger preprocessing such as PCA whitening, especially given the Evo1 result?
4. Can the authors provide broader evidence that mean-centering improves feature usefulness/interpretability, beyond reconstruction and probing?

**Limitations:**

Yes

**Strengths And Weaknesses:**

**Strengths:** clear mechanism, strong empirical correlation, synthetic causal evidence, and a simple practical fix with low cost. The paper also explains why recovery is slow and why AuxK helps only in some regimes.

**Weaknesses:** the theory is strongest for dead-by-ReLU; dead-by-TopK is supported mainly empirically. The evaluation of feature usefulness is still limited, since it mainly uses reconstruction/probe metrics rather than broader interpretability evidence. The paper also admits γ does not explain everything (e.g., Evo1, anisotropy).

---

> ### Author Rebuttal · Authors · 2026-03-31
>
> Thank you for the detailed and constructive review. Your suggestions around TopK theory, robustness checks, and feature quality evaluation have significantly strengthened the paper. We have run several new experiments in response, and the resulting figures and tables are available at [https://anonymous.4open.science/r/sae](https://anonymous.4open.science/r/sae). We address each point below.
>
> ## Theoretical understanding of dead-by-TopK
>
> Dead-by-ReLU and dead-by-TopK are the same mechanism with different thresholds. The ReLU derivation asks whether a feature's shift is too negative for any input to push it above zero. For TopK the bar is higher: a feature must beat the k-th ranked feature. At high γ the shifts dominate, so that bar is approximately fixed at the (1 − k/n) percentile of the shift distribution, giving P(dead, TopK) ≈ Φ(t_k − C/γ), with the ReLU formula as the special case t_k = 0. topk.pdf shows this theory curve overlaid on both synthetic and real data. We will include the full derivation in the revision.
>
> ## Robustness to SAE width, sparsity k, optimizer, training budget
>
> We have added additional results to show that γ holds across all four axes:
>
> - `width.pdf` sweeps width from 10× to 100× and k from 5 to 100. γ predicts death rates across all combinations for both pathways. The complementary experiment (fixed μ, varying σ, and vice versa) confirms γ is the operative quantity, not μ or σ individually.
> - `lr.pdf` sweeps learning rate from 1e-5 to 3e-3. Baseline TopK has 26–95% dead features across rates — a wide range that makes hyperparameter tuning unpredictable. Mean-centering drops this to 1–6% across the full range, both eliminating most death and making results stable across learning rates. The same pattern holds for MMCS. Figure 6 in the submission shows mean-centering also reduces sensitivity to training budget: it converges faster by skipping bias learning, so practitioners don't need extended training runs to recover from outlier-induced death.
>
> ## When to use mean-centering vs. PCA whitening
>
> The underlying geometry has two independent failure modes, and we will add a section to the paper making this diagnostic clearer.
>
> - **Large mean offsets** (high γ) predetermine feature fate at initialization; mean-centering removes this.
> - **Eigenvalue concentration** is a separate problem: when a few PCs carry most of the variance, features aligned with those PCs always win the TopK competition regardless of mean offsets.
>
> Evo1 is the clearest case: 4 of 4096 PCs carry 99% of variance, mean-centering alone still leaves 67% dead, and partial eigenvalue equalization reaches 0% at α = 0.75 (`pca.pdf`). A subtlety: eigenvalue concentration has two sources. On DINOv3 and ModernBERT, per-token norm variation inflates top eigenvalues artificially; LayerNorm removes this and mean-centering then works. On Evo1, ESM3, ESM2-3B, and ProGen, activations genuinely occupy a small subspace, and only equalization helps. Both failure modes are diagnosable from a single forward pass. The preprocess table (`preprocess.pdf`) gives the recommended fix for all 19 models.
>
> ## Feature usefulness / interpretability beyond reconstruction
>
> We have added Monosemanticity Score evaluations ([Pach et al., 2025](https://openreview.net/forum?id=DaNnkQJSQf)) on DINOv3 and ESM3, measuring whether individual features respond to semantically coherent inputs via pairwise embedding similarity with independent models (CLIP-ViT and ESM2 respectively). This is methodologically independent of both reconstruction error and linear probing.
>
> `MS_score.pdf` shows the full distributions. MC SAEs have far more features with meaningful monosemanticity — baseline curves drop to zero early while MC curves maintain nonzero scores across most of the dictionary. Even restricting to alive features only, MC features score higher at every quantile in both models. Raw neuron activations (no SAE baseline) are flat near zero, confirming SAEs perform real decomposition and MC SAEs do it more effectively.
>
> Together with MMCS on synthetic data (Sharkey et al., 2022) and per-feature biological concept alignment on ESM3 (Simon & Zou, 2025), these evaluations show across three domains that mean-centering produces features that are more interpretable.

---

> > ### Author Rebuttal · Reviewer_VFKD · 2026-04-03
> >
> > Thanks for the authors’ rebuttal, I will maintain my original score.

---

> > > ### Author Response · Authors · 2026-04-07
> > >
> > > Thank you for confirming that your concerns have been fully addressed. The new experiments we ran in response to your review have meaningfully strengthened the paper.
> > >
> > > As all of your concerns have been resolved, we would very much appreciate it if you would consider updating your score to reflect this.

---

### Official Review · Reviewer_zq6f · 2026-03-17

**Soundness:** 3
**Presentation:** 2
**Significance:** 3
**Originality:** 3
**Overall Recommendation:** 4
**Confidence:** 4

**Summary:**

This paper addresses the problem of dead features in SAE. Particularly, the paper identifies dimension-level activation outliers as a cause for dead features, introduces a metric that predicts feature death, describes dead-by-TopK and dead-by-ReLU as two mechanisms that activation outliers can lead to feature deaths, and validates the hypotheses with both synthetic and empirical experiments. This paper also provides practical implications for when mean-centering is necessary for SAE training.

**Compliance With Llm Reviewing Policy:**

Affirmed.

**Final Justification:**

The authors have addressed the concerns I raised during the rebuttal and follow-up, so I maintain my initial, positive assessment of the work.

**Key Questions For Authors:**

1. Are the results in Table 1 and Figure 5 from the synthetic settings or the real-world settings? If they are from the synthetic settings, do the results in Table 1 and Figure 5 hold for activations from real-world models? This is my main concern.
2. Why do we need to compute $\gamma$ to decide whether mean-centering is needed? Why not simply perform mean-centering for every SAE training?
3. For ESM2-3B and ESM2-650M, does mean-centering help improve the F1 score for protein concept detection? For other models of other domains, how does mean-centering impact the SAE performance of concept detection?
4. With mean-centering, are the SAE features more monosemantic compared to an SAE with more dead features?

**Limitations:**

Yes

**Strengths And Weaknesses:**

**Strengths**
1. This paper provides a new perspective on the cause of dead features. That is, dead features can be caused by the activations' dimension-level distributions. This is important because in this case dead features are the artifact of poor initialization and training, instead of explained by an activation space inherently having few concepts encoded. I encourage the authors to include this distinction in their discussion.
2. The perspective proposed by the paper is supported with theoretical intuitions (Sections 4.1-4.2), synthetic experiments (Section 5), and empirical validation (Section 5-6).
3. The authors' theoretical exposition focuses on dead features at random initialization. However, the authors also empirically verified that training cannot fully resolve the problem of dead features (Section 5). This provides empirical evidence that mean-centering is still necessary in practice.

**Weaknesses**
1. It is unclear whether the results in Table 1 and Figure 5 are from the synthetic activations or real-world models' activations.
2. The practical implication of mean-centering is only evaluated on ESM-3, with increased F1 score for protein concept detection with mean-centering. I encourage the authors to include more concept detection results on real-world models to demonstrate the practical impact of their analysis.
3. An analysis on whether mean-centering lead to more monosemantic features is lacking. This is important because in the Introduction the authors mention that, with a large number of dead features, the surviving features can be forced into superposition.

---

> ### Author Rebuttal · Authors · 2026-03-31
>
> Thank you for the detailed and encouraging review. We're glad the mechanism and diagnostic landed clearly, and your questions about real-data dynamics, concept sparsity, and monosemanticity have pushed us to add experiments that we believe make the paper substantially stronger. All new figures and tables are available at [https://anonymous.4open.science/r/sae](https://anonymous.4open.science/r/sae). We address each point below.
>
> ## The two-phase dynamics hold on real data (Table 1 / Figure 5)
>
> Table 1 and Figure 5 are from synthetic settings, yet the same dynamics hold on real models. dynamics.pdf shows the dead-by-TopK / dead-by-ReLU breakdown for three real models (AlphaFold3, Stable Diffusion 3.5, ESM3). All three exhibit the same pattern: sudden TopK revival during the collapse window, conversion of features to dead-by-ReLU, then a plateau. The bottom row shows the bottleneck directly: after 100K steps, bias has captured less than 0.3% of the activation mean in every case. These models have large mean offsets, so the required bias shift is correspondingly large, and bias learning is too slow to make meaningful progress within practical training budgets. Until the bias catches up, dead-by-ReLU features have no mechanism to revive, which is why they plateau.
>
> These dynamics were actually first observed in real activations — the synthetic setting lets us isolate outliers from other distributional factors, but the phenomenon is not an artifact of the controlled setup. We will make the synthetic/real distinction more explicit in the text and add this figure to the supplement.
>
> ## Why compute γ? Why not always mean-center?
>
> Mean-centering is a good default and we'd recommend it. γ is more useful for understanding *why* models differ than for deciding *whether* to center. It quantifies how much activation geometry predetermines feature fate, explains why identical SAE configs succeed on some models and fail on others, and predicts death rates from first principles.
>
> That said, investigating why mean-centering doesn't solve Evo1 led us to identify two additional failure modes:
>
> - **High per-token norm variation** (DINOv3, ModernBERT): needs LayerNorm before centering; without it, per-dimension means mix in token-level scale variation and centering doesn't accurately remove the constant offset.
> - **Low-rank activations** (Evo1, ESM3): needs PCA whitening regardless.
>
> `preprocess.pdf` shows the diagnostic and recommended fix for all 19 models. So γ and the eigenvalue spectrum do end up telling you which preprocessing to use, which we will include in the paper revision, but we think the more important contribution is understanding *why* these interventions work.
>
>
> ## Monosemanticity and concept detection on other models
>
> We computed Monosemanticity Scores (Pach et al., 2025) on DINOv3 and ESM3, measuring whether individual features respond to semantically coherent inputs via pairwise embedding similarity with independent models (CLIP-ViT and ESM2 respectively). `MS_score.pdf` shows the full distributions. Two things stand out:
>
> 1. MC SAEs have far more features with meaningful monosemanticity — baseline curves drop to zero early while MC curves maintain nonzero scores across most of the dictionary.
> 2. This isn't just an artifact of having more alive features: restricting to alive features only, MC features are more monosemantic at every quantile.
>
> Raw neuron activations (no SAE baseline) are flat near zero, confirming SAEs perform real decomposition and MC SAEs do it more effectively. This is consistent with your point — when most features are dead, survivors are forced into superposition and decompose less cleanly.
>
> For concept detection on ESM2-3B and ESM2-650M: Swiss-Prot alignment experiments are in progress and will be included in the revision.
>
>
> ## Distinguishing geometric death from concept sparsity
>
> This is an important distinction that we will make explicit in the revision. Geometric death (our focus) is predetermined at initialization by alignment with the activation mean. If death were instead caused by the dictionary being larger than the number of true concepts, the signatures would differ:
>
> 1. It would emerge during training as unused features decay, not be present at init.
> 2. Models with more dead features should have better reconstruction-sparsity tradeoffs, since the alive features would be exactly the right ones. We see the opposite.
> 3. Surviving features should be more interpretable, since they'd map onto real concepts without overcrowding. Again the opposite: our MS evaluations show alive features in high-death SAEs are less monosemantic than alive features in MC SAEs.
>
> Geometric death wastes capacity and forces survivors into superposition; the data don't look like a dictionary that's simply too large.

---

> > ### Author Rebuttal · Reviewer_zq6f · 2026-04-04
> >
> > - Thank you for providing the dynamics plots on real-world models. I encourage the authors to show them in the main text as well to validate the results on synthetic models. W1 and Q1 are addressed.
> > - Q2 is sufficiently addressed with an explanation on how to use $\gamma$ in practice.
> > - W3 and Q4 are sufficiently addressed with the empirical results on monosemanticity scores, showing that mean centering can lead to improvement.
> > - W2 and Q3 remain unresolved. I look forward to the concept detection results on ESM2-3B and ESM2-650M.

---

> > > ### Author Response · Authors · 2026-04-07
> > >
> > > Thank you for the engagement throughout this discussion — your questions pushed us to run experiments that meaningfully strengthen the paper.
> > >
> > > * Regarding your suggestion to incorporate the real-model dynamics plots into the main text (W1/Q1): we agree this improves the paper and will include them in the revision to validate the synthetic results directly alongside real-world models.
> > > * Regarding W2/Q3, the remaining unresolved concern: we have now completed the Swiss-Prot concept detection experiments on ESM2-3B and ESM2-650M (selected as high-γ layers where geometric death is most severe). Mean-centering substantially improves concept recovery: concepts detected at F1 > 0.7 increase from 5 to 109 on ESM2-3B layer 3 and from 7 to 49 on ESM2-650M layer 2. This is consistent with our earlier findings on ESM3. A new plot capturing this result is available at https://anonymous.4open.science/r/sae/esm2_dead_comparison.pdf.
> > >
> > > ----
> > >
> > > Together with the monosemanticity score improvements you noted as resolved (W3/Q4) and the practical γ guidance (Q2), the paper now shows across multiple models and evaluation methodologies that mean-centering improves not just dictionary capacity but the quality and semantic coherence of individual features. We believe all of your concerns have now been addressed with new empirical evidence. If you agree, we would very much appreciate it if you would consider updating your score.

---

### Decision · Program_Chairs · 2026-04-30

**Decision:**

Accept (regular)

**Comment:**

This paper studies why sparse autoencoders often develop dead features, and argues that this failure can be substantially determined at initialization by the geometry of the activation distribution, rather than arising only from later training dynamics. More specifically, it identifies dimension-level activation outliers as the main driver of the feature-death mechanism studied here, introduces a simple diagnostic to predict death rates, distinguishes two pathways (dead-by-ReLU and dead-by-TopK) and shows that mean-centering at initialization can largely eliminate the resulting death. The reviewers generally found the paper clear, well motivated and practically useful, and several regarded the mechanism as convincing because it is supported by theory, synthetic causal experiments and substantial empirical evidence across multiple model families and modalities. The main reservations were limited rather than fundamental. The theoretical account was seen as strongest for dead-by-ReLU, with dead-by-TopK initially more empirically grounded. Some reviewers also wanted broader evidence that the intervention improves downstream feature quality and interpretability, beyond reconstruction and probing, and clearer guidance on when mean-centering is enough versus when stronger preprocessing, such as LayerNorm or PCA whitening, is needed. Overall, however, the discussion was positive: the paper was seen as offering a compelling new explanation for SAE feature death, together with a simple diagnostic and a practical remedy, even if some parts of the theory and downstream validation remain more developed than others.